# COVID-19: Spatial analysis of hospital case-fatality rate in France

Marc Souris[ID][1]*, Jean-Paul Gonzalez[ID][2,3,4]

**1** UMR Unité des Virus Émergents (UVE Aix-Marseille Univ-IRD 190-Inserm 1207-IHU Méditerranée Infection), Marseille, France, **2** Department of Microbiology & Immunology, School of Medicine, Georgetown University, Washington, DC, United States of America, **3** Commonwealth Trade Partners Inc., Alexandria, VA, United States of America, **4** Centaurus Biotech LLC, Manassas, VA, United States of America

* marc.souris@ird.fr

**Data Availability Statement:** The data underlying the results presented in the study are available from www.data.gouv.fr/fr/datasets/donnees-hospitalieres-relatives-a-lepidemie-de-covid-19. The authors of the present study had no special

## Abstract

When the population risk factors and reporting systems are similar, the assessment of the case-fatality (or lethality) rate (ratio of cases to deaths) represents a perfect tool for analyzing, understanding and improving the overall efficiency of the health system. The objective of this article is to estimate the influence of the hospital care system on lethality in metropolitan France during the inception of the COVID-19 epidemic, by analyzing the spatial variability of the hospital case-fatality rate (CFR) between French districts. In theory, the hospital age-standardized CFR should not display significant differences between districts, since hospital lethality depends on the virulence of the pathogen (the SARS-CoV-2 virus), the vulnerability of the population (mainly age-related), the healthcare system quality, and cases and deaths definition and the recording accuracy. We analyzed hospital data on COVID-19 hospitalizations, severity (admission to intensive care units for reanimation or endotracheal intubation) and mortality, from March 19 to May 8 corresponding to the first French lockdown. All rates were age-standardized to eliminate differences in districts age structure. The results show that the higher case-fatality rates observed by districts are mostly related to the level of morbidity. Time analysis shows that the case-fatality rate has decreased over time, globally and in almost all districts, showing an improvement in the management of severe patients during the epidemic. In conclusion, it appears that during the first critical phase of COVID-19 ramping epidemic in metropolitan France, the higher case-fatality rates were generally related to the higher level of hospitalization, then potentially related to the overload of healthcare system. Also, low hospitalization with high case-fatality rates were mostly found in districts with low population density, and could due to some limitation of the local healthcare access. However, the magnitude of this increase of case-fatality rate represents less than 10 per cent of the average case-fatality rate, and this variation is small compared to much greater variation across countries reported in the literature.

privileges in accessing these datasets which other interested researchers would not have.

**Funding:** The authors received no specific funding for this work.

**Competing interests:** The authors have declared that no competing interests exist.

## Introduction

Since the beginning of the epidemic, the lethality rate, or case-fatality rate (CFR) of COVID-19 and the differences between countries have been the subject of many questions about national pandemic response policies and patient treatment. Most studies on the lethality of COVID-19 seek to estimate the true lethality of the disease, an issue that has been addressed since the beginning of the epidemic [1–4].

The case-fatality rate is the ratio between the number of deaths due to the disease and the number of closed cases (i.e. recovered or dead). It is estimated by the healthcare system based on the reporting of these two values. The case-fatality rate should not be confused with the mortality rate, which is the ratio of the number of deaths to the total population, or also with the morbidity rate, which is the ratio of the number of cases to the total population. Mortality and morbidity rates depend on the extent of disease in a population, unlike case-fatality rates [5].

The case-fatality rate of a disease in a population is an index of severity of the disease in that population, and of the capacity of the healthcare system to reduce mortality. In principle, this allows to compare the effectiveness of healthcare systems across regions or countries.

The aim of this article is to analyze the effectiveness of the healthcare system in France in the context of the COVID-19 epidemic. This study also discusses the differences and the comparison on case-fatality rates published by the international agency by country (May 2020).

Lethality depends on the virulence of the virus but, unlike morbidity, it does not depend on its contagiousness. Virulence comes from the reproductive capacity of the virus in the cell, its capacity for cellular degradation, and its ability to induce or not an innate or specific immune response. Virulence is of purely biological origin and once the virus has entered the target cell where it will cause its pathogenic effect does no longer depends on environmental conditions outside the host. Virulence is independent of the host population but may change over time and space if there is a risk of natural mutation/selection of the pathogen [6–8]. Contagiousness characterizes the biological capacity of the virus to reach the target cell system of its host, and the ability to be transmitted from one individual to another. The efficiency of transmission depends largely on environmental conditions (e.g., climate, urbanization, population density, mobility), which can vary greatly from one country to another.

In addition to the virulence of the virus, the case-fatality rate depends on viral load, on biological risk factors and on population vulnerability (age structure, genetic factors, prevalence of co-morbidities, healthcare accessibility, etc.) as well as other factors related to the health system (equipment, capacity, staff, management, care of patients, effectiveness of therapies, patient management in a critical phase of the disease), and factors related to the detection and registration system for cases and deaths (clinical cases definition, detection, surveillance systems, case and death reporting). The evaluation of the case-fatality rate normally requires the detection and counting of all infected persons, irrespective of their level of symptoms (i.e. disease severity).

When the population risk factors and reporting systems are identical, case-fatality rate evaluation represents an excellent tool for analyzing, understanding and improving the overall performance of the health system, particularly at the level of hospital units. Studying the magnitude of differences in case-fatality rates between units also makes it possible to assess the impact of the quality of the health system on case-fatality.

There are large differences in the case-fatality rates of COVID-19 published by country (Table 1) or calculated directly from WHO data. These rates vary considerably, from less than 0.02 (Thailand, Australia, Chile) to more than 0.15 (France, Belgium, UK), with a mean at 0.04 and a standard deviation of 0.045 (WHO, May 8, 2020, Fig 1).

**Table 1. Case-fatality rates and characteristics of countrywide population vulnerability to COVID-19.**

| Country | CFR (%)* | Population > 65-year-old (%)** | Hospital bed number per 100,000 people*** |
|---|---|---|---|
| France | 19.00 | 20.02 | 621 |
| Belgium | 16.42 | 18.80 | 623 |
| UK | 14.94 | 18.40 | 273 |
| Italy | 13.84 | 22.75 | 331 |
| Netherlands | 12.59 | 19.20 | 466 |
| Sweden | 12.29 | 20.10 | 254 |
| Spain | 11.73 | 19.38 | 297 |
| Iran | 6.31 | 6.18 | 150 |
| USA | 5.40 | 15.80 | 290 |
| Greece | 5.52 | 21.65 | 424 |
| China | 5.50 | 10.92 | 420 |
| Switzerland | 5.01 | 18.62 | 470 |
| Germany | 4.28 | 21.46 | 823 |
| Portugal | 4.15 | 21.95 | 332 |
| Austria | 3.88 | 19.02 | 760 |
| Tanzania | 3.75 | 2.60 | 70 |
| Japan | 3.56 | 27.58 | 1,340 |
| India | 3.36 | 6.18 | 70 |
| Czech Republic | 3.28 | 19.42 | 645 |
| Turkey | 2.72 | 8.48 | 270 |
| Norway | 2.62 | 17.05 | 390 |
| Uruguay | 2.53 | 14.81 | 280 |
| South Korea | 2.36 | 14.42 | 1150 |
| Thailand | 1.83 | 11.90 | 210 |
| Slovakia | 1.74 | 15.63 | 579 |
| Australia | 1.41 | 15.65 | 380 |
| Chile | 1.21 | 11.53 | 220 |

*Lethality rate for Covid-19 reported among selected countries from WHO, as for May 8, 2020 (https://covid19.who.int/)

**population percentage over 65-year-old by World Bank, 2016

***Hospital beds per 100,000 people by World Bank, 2013, Eurostat, 2014.

In Europe, the characteristics of populations (in terms of risk factor for COVID-19) and health systems are comparable, but the definition, detection and reporting of cases and causes of death can differ greatly from one country to another. Some countries conducted significantly more detection tests and hospitalizations than others (Table 2), resulting in differences in the protocols for patient management. The rate of testing performed (policy) and mortality rates (reporting) vary mainly according to the geographical extent of the epidemic within each country.

The virulence of the COVID-19 pathogen (SARS-CoV-2 virus) is assumed to be identical in all countries. In order to compare case-fatality rates across regions or countries (and thus analyze the effectiveness of healthcare systems), it is necessary, when calculating rates, to standardize population-related risk factors and to use the same definitions and enumeration methods to record cases and deaths. This is not the case for the current pandemic and discrepancies exist among the country systems.

The principal objective of this article is not to estimate the actual lethality of COVID-19 in France based on the rates published by the health authorities, but to estimate the influence of the healthcare system on lethality by analyzing the spatial variability of the hospital case-fatality

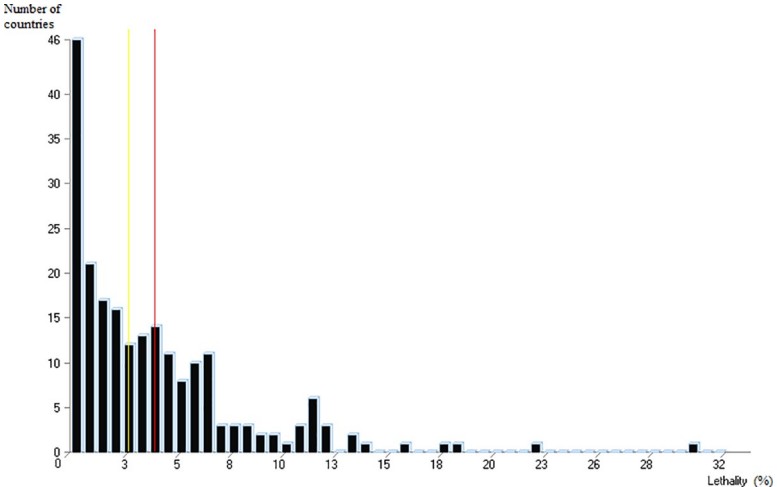

**Fig 1. Distribution of case-fatality rates calculated by country, based on case and death data from WHO (WHO, May 8, 2020).**

rate (confirmed hospitalized cases and hospital deaths) in metropolitan France between districts (i.e. French *départements*). This analysis, limited to metropolitan France, is possible while it remains within the framework of the same system for defining and counting cases and deaths. We thus assume that this system of definition and enumeration was identical throughout France during the period (19 March to 8 May) corresponding to the first wave (inception) of the COVID-19 epidemic in France. Therefore the study focuses on the extent of spatial differences in the case-fatality rate in metropolitan France, and enables to highlight the relative differences between districts, as well as to analyze the causes independently of the system of definition and enumeration of cases and deaths, and also independently of the main biological risk factor of severity (age) after standardization on this factor.

The other objective is to analyze the case-fatality rate observed in metropolitan France with the one calculated for other countries: the variability of the case-fatality rate between French

**Table 2. SARS-Cov-2 detection, COVID-19 morbidity and mortality rates by selected country.**

| Country | rtPCR Test number per 1,000* | Morbidity rate (cases per million) | Mortality rate (death per million) |
|---|---|---|---|
| Belgium | 44 | 4488 | 735 |
| Spain | 41 | 5494 | 558 |
| Italy | 39 | 3570 | 495 |
| Germany | 33 | 2022 | 88 |
| Australia | 29 | 271 | 4 |
| United States | 25 | 3906 | 232 |
| United Kingdom | 22 | 3045 | 451 |
| France | 21 | 2678 | 398 |
| Turkey | 15 | 1586 | 43 |
| Netherlands | 14 | 2438 | 309 |
| South Korea | 12 | 211 | 5 |
| Iran | 6 | 1246 | 78 |
| Japan | 1.5 | 122 | 4.5 |
| India | 1 | 41 | 1.4 |
| Indonesia | 0.5 | 48 | 3.5 |

Source: Worldometer, May 8, 2020.

districts (mainly due to patient management in the acute epidemic phase) allow us to estimate whether this observed variability can explain the significant differences in case-fatality rates observed between France and other countries.

## Data and methods

### Data

This study is based on daily hospitalization and death declaration data by district in France and is accessible on the "Santé Publique France" website. (www.data.gouv.fr/fr/datasets/ donnees-hospitalieres-relatives-a-lepidemie-de-covid-19) from March 19 to May 8, 2020, corresponding to 50 days lockdown (i.e. quarantine) and the spread of the COVID-19 epidemic in France (Fig 2). We used data on the distribution of hospitalized COVID-19 cases according to age group (10-year age group) (Santé Publique France). We also obtain demographic data by districts (source: population by age, INSEE, 2020, www.insee.fr), healthcare system (number of hospital beds) and some health status rates (hypertension, coronary disease, acute respiratory disease, diabetes) by districts (source: Ecosanté France, www.ecosante.fr, 2016),

This analysis was carried out on the 96 districts of metropolitan France (Fig 3), while the French overseas districts and territories were excluded from the analysis for reasons of spatial analysis and mapping. The data were integrated into a geographic information system (Sav-GIS, ww.savgis.org) for analysis and mapping.

Focusing only on confirmed hospitalized cases and deaths in hospital, the case-fatality rates calculated in this article for France refer to the hospital case-fatality rates and do not represent the absolute national case-fatality that should include all positive cases and deaths related to COVID-19. Given the estimated high number of non-severe and asymptomatic forms (which a fortiori do not cause deaths)—it is estimated that only 2.6% of infected persons were hospitalized [9]—this overall lethality is necessarily much lower than hospital lethality, but it will be accurately calculated only at the end of the epidemic when the total number of positive cases (i.e. seroprevalence survey) will be available and the total number of deaths outside hospital due to COVID-19 will be accurately assessed.

All identified and hospitalized cases were tested positive (by rtPCR). All deaths counted were COVID-19 associated. As of May 8, 2020, not all hospitalized cases are closed since the epidemic is still ongoing: deaths counted at the beginning of the study period correspond to cases hospitalized but were not included in the study, and cases counted at the end of the period were not closed and no deaths from these cases were included in the study.

The hospital healthcare system in France offer a mean of 621 Hospital beds per 100,000 people. The hospital healthcare system is fairly evenly distributed throughout the metropolitan territory, as shown in Table 3 for regions (www.ecosante.fr).

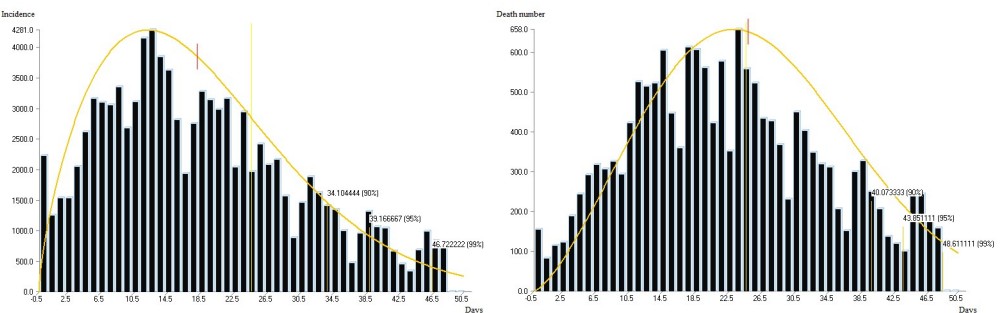

**Fig 2.** COVID-19 timeline of hospitalizations (left) and death (right), March, 19 to May, 8 2020, France.

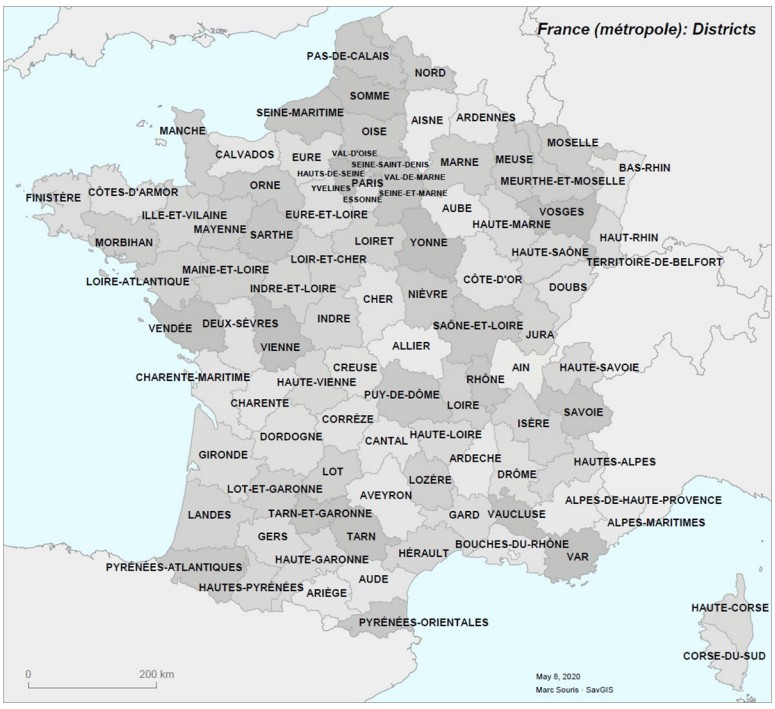

**Fig 3. The 96 districts of metropolitan France.**

## Methods

**Age standardization.** Since age is a major risk factor for death for COVID-19 patients, to compare hospitalization rates, mortality rates, or case-fatality rates between districts it's important to eliminates the differences between districts due to differences in age structure of the population of the districts. We performed an indirect standardization [10] by calculating the rate (hospitalization rate and hospital lethality rate) per ten-years age group for the whole of France. We also estimated lethality rate for the whole of France excluding areas that have been under hospital stress (Ile-de-France and Grand-Est), in order to evaluate hospital case-fatality rates per age group independently of possible excess mortality due to the saturation of healthcare systems in certain districts. Classically, $E_i$ (expected number of deaths in district $i$) is the sum over the different age groups of the district's population in the age group multiplied by the hospital case-fatality rate of the age group:

$$E_i = \sum_{age\ class\ a} P_{i,a} \times T_a$$

where $P_{i,a}$ is the population of the age group "a" of the district $i$ and $T_a$ is the hospital case-fatality rate of the age group calculated over the whole country.

**Standardized case-fatality rate.** The Standardized Lethality Ratio (SLR) is equal to the ratio between the observed death toll $Oi$ and the expected death toll $Ei$. The standardized hospital case-fatality rate $LS_i$ for each district is equal to the average rate for France multiplied by the district's SLR. The significance of the SLR (H0: SLR = 1, no statistically significant over—or under-fatality compared to the expected value) is evaluated for each district by a Breslow & Day test [11, 12] in order to account for the statistical significance of the differences in these ratios between districts. In order to take into account the problem of multiple testing [13] when the null hypothesis concerns the whole territory ("is there a district for which the H0

**Table 3. Hospital bed number per 100,000 people, per regions in France.**

| Region | Name | Hospital bed number per 100,000 people |
|---|---|---|
| France | Whole country | 621 |
| 11 | Île de France | 547 |
| 24 | Centre-Val de Loire | 634 |
| 27 | Bourgogne-Franche-Comté | 711 |
| 28 | Normandie | 621 |
| 32 | Hauts-de-France | 634 |
| 44 | Grand Est | 642 |
| 52 | Pays de la Loire | 569 |
| 53 | Bretagne | 651 |
| 75 | Nouvelle Aquitaine | 686 |
| 76 | Occitanie | 691 |
| 84 | Auvergne-Rhône-Alpes | 659 |
| 93 | Provence-Alpes-Côte d'Azur | 778 |
| 94 | Corse | 678 |

hypothesis is rejected?"), the risk of error was set at 0.0005 in order to apply a Bonferonni correction (which is very conservative when data is spatially autocorrelated). When the study is conducted for a single district ("Is the SLR for this district different from 1?"), the risk of error is set at 0.05. All tests are bilateral.

Besides age, another risk factors for severity and mortality have been observed for COVID-19, including hypertension, cardiovascular disease history, diabetes, obesity. Even if these risk factors are often related to age, to detect another risk factor for lethality (as healthcare saturation) using districts comparison, it would be useful to standardize on these factors to. Unfortunately, apart for age, we do not have the necessary data to evaluate the global case-fatality rates due to COVID-19 for these risk factors or combination of factors. To overcome this difficulty, we analyzed relationships between CFR and health risk factors rates (diabetes, coronary disease, hypertension, respiratory syndrome), but these correlations should be used with caution due to possible ecological fallacy (they should be done at individual level, but these data are not available).

**Cartography.** The values of morbidity rates, mortality rates, non-standardized case-fatality rates, SLRs and their significance were mapped by districts. In order to map case-fatality rates independently of the rate calculation variability, due to differences in hospitalization numbers between districts, rates adjusted with empirical Bayesian estimator (EBE) were calculated [14]. All statistical and spatial analysis and mapping were carried out using the SavGIS geographic information system v. 9.15 (www.savgis.org).

**Study of statistical correlations.** In order to estimate whether possible congestion in the healthcare system influenced hospital case-fatality rate, we studied the correlations between hospitalization and hospital lethality by district over the entire period. Inpatient severity was estimated using the ICU rate (number of patients admitted in intensive care /number of inpatients). We analyzed also relationship between standardized CFR and healthcare system capacity (bed number per people).

## Results

For the whole of France (data from 19 March to 8 May 2020), the hospital case-fatality rate was 0.174. The average hospital case-fatality rates by age group, used in indirect standardization, was calculated from French Public Health data by age class published on 8 May 2020 (Table 4).

**Table 4. Hospitalization, mortality and case-fatality rates by age group in metropolitan France (2020).**

| Age class | Population (%) | Hospitalization | | | | | | |
|---|---|---|---|---|---|---|---|---|
| | | Hospitalized | | Death | | CFR | | |
| | | Total | % | Total | % | Whole country | Without #11 & 44 regions | #11 & 44 regions only |
| 0–9 | 11.43 | 617 | 0.65 | 2 | 0.01 | 0.0032 | 0 | 0.0049 |
| 10–19 | 12.27 | 463 | 0.49 | 3 | 0.02 | 0.0065 | 0 | 0.0125 |
| 20–29 | 11.15 | 2253 | 2.4 | 19 | 0.11 | 0.0084 | 0.0051 | 0.0121 |
| 30–39 | 12.36 | 4325 | 4.5 | 72 | 0.44 | 0.0166 | 0.0088 | 0.0229 |
| 40–49 | 12.80 | 7001 | 7.4 | 203 | 1.24 | 0.0290 | 0.0183 | 0.0372 |
| 50–59 | 13.10 | 12130 | 12.9 | 747 | 4.56 | 0.0616 | 0.0435 | 0.0751 |
| 60–69 | 12.01 | 16115 | 17.1 | 1934 | 11.81 | 0.1200 | 0.0988 | 0.1361 |
| 70–79 | 8.57 | 18812 | 20.0 | 3723 | 22.73 | 0.1979 | 0.1666 | 0.2244 |
| 80–89 | 4.91 | 21625 | 22.9 | 6192 | 37.80 | 0.2863 | 0.2688 | 0.3033 |
| 90+ | 1.38 | 10897 | 11.5 | 3485 | 21.28 | 0.3198 | 0.3047 | 0.3346 |
| All ages | 100 | 94238 | 100 | 16380 | 100 | 0.1738 | 0.1597 | 0.1858 |

(Region 11 = Ile-de-France; region 14 = Grand-Est).

The epidemic in France (data from 19 March to 8 May 2020) presents a very uneven spatial distribution of hospitalizations. It has spread throughout the country at low level, with a mean hospitalization rate of 1.27 per 1,000 (median at 0.88) and a standard deviation of 1.03, except in some regions where transmission has been much more active, such as the Ile-de-France region (mean hospitalization rate 3.01 per 1,000), or the Grand-Est region (mean hospitalization rate 3.2 per 1,000). The mean hospitalization rate varies from 0.19 per 1,000 (Tarn-et-Garonne) to 4.75 (Territoire de Belfort) (Fig 4).

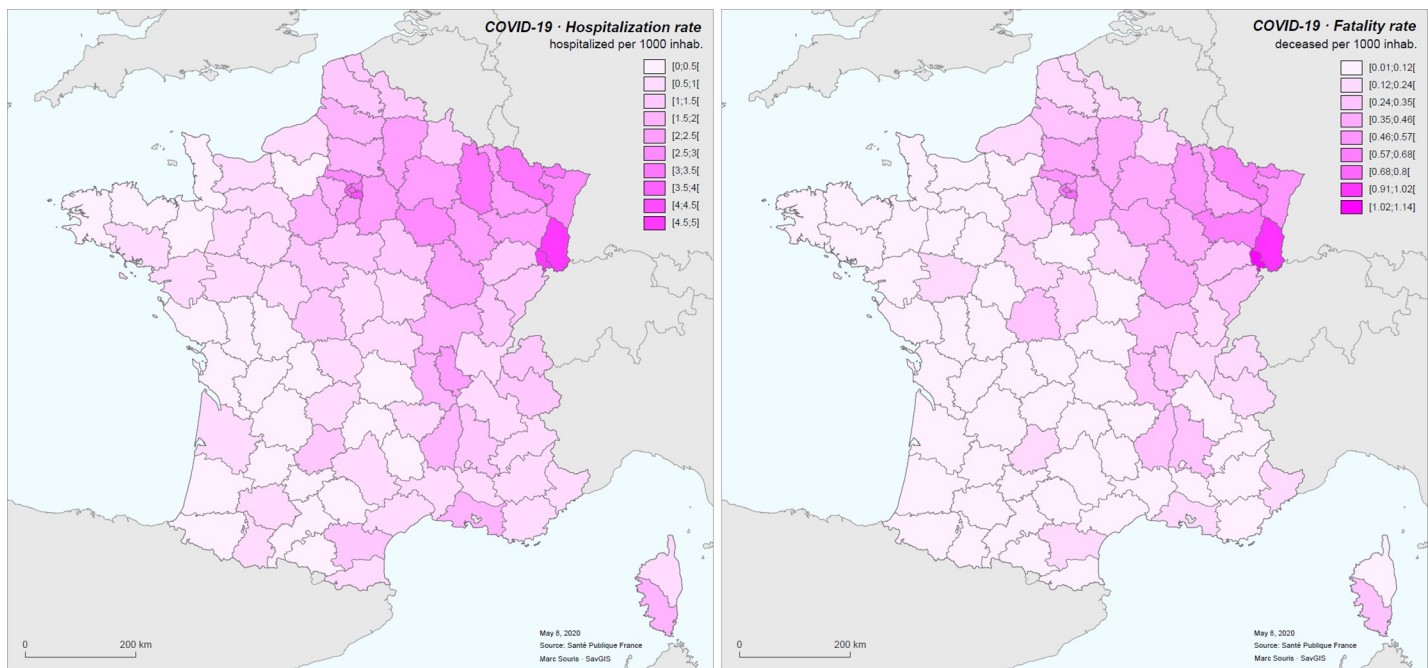

**Fig 4.** Hospitalization rate (left) and hospital mortality rate (per 1000 inhabitants) (right) for COVID-19 in metropolitan France for the period March 19 to May 8, 2020.

The hospital mortality rate (not age-standardized) has the same spatial distribution. It varies from 0.01 per 1,000 (Tarn-et-Garonne) to 1.13 per 1,000 (Territoire de Belfort), with a mean of 0.21 (median 0.12) and a standard deviation of 0.21.

Such relative differences in the circulation of the disease reported by districts in French metropolitan area clearly appears in the cartogram displayed below (Fig 5).

The observed gross hospital case-fatality rate calculated by district showed a mean of 0.153 with a standard deviation of 0.045. It varies between 0.046 (Ariège) and 0.258 (Vosges). Without Ile-de-France and Grand-Est regions, the mean gross case-fatality rate per district is 0.149 with a standard deviation of 0.048 (Fig 6).

Case-fatality rates have evolved over the period of the study. Here we represent non-standardized rates calculated for three periods: March 19 to 31; April 1 to 12; April 13 to 25. Rates are calculated with a 12-day time lag between cases and deaths (Fig 7).

The standardized case-fatality rate per district is between 0.044 (Ariège, SLR = 0.25) and 0.255 (Vosges, SLR = 1.48), with the mean at 0.151 and the median at 0.153. If the Ile-de-France and Grand-Est regions are excluded from the calculation of age-specific case-fatality rates, the CFR is between 0.044 and 0.251, with the mean at 0.144 and a median at 0.147 (SLR between 0.28 and 1.67, mean at 0.99 and median at 1) (Fig 8). In the following, we will consider only the standardized case-fatality rates calculated with age-specific case-fatality rates that do not take into account the Ile-de-France and Grand-Est regions.

The spatial distribution of standardized morbidity rate (hospitalized cases) shows a significant spatial autocorrelation (Moran index: 1.54, p-value $< 10^{-6}$), and this is expected for an infectious disease [14, 15]. The case-fatality rate shows also significant spatial autocorrelation (Moran index: 0.29, p-value $< 0.000007$), and this is no expected. The analysis of the clusters clearly shows a

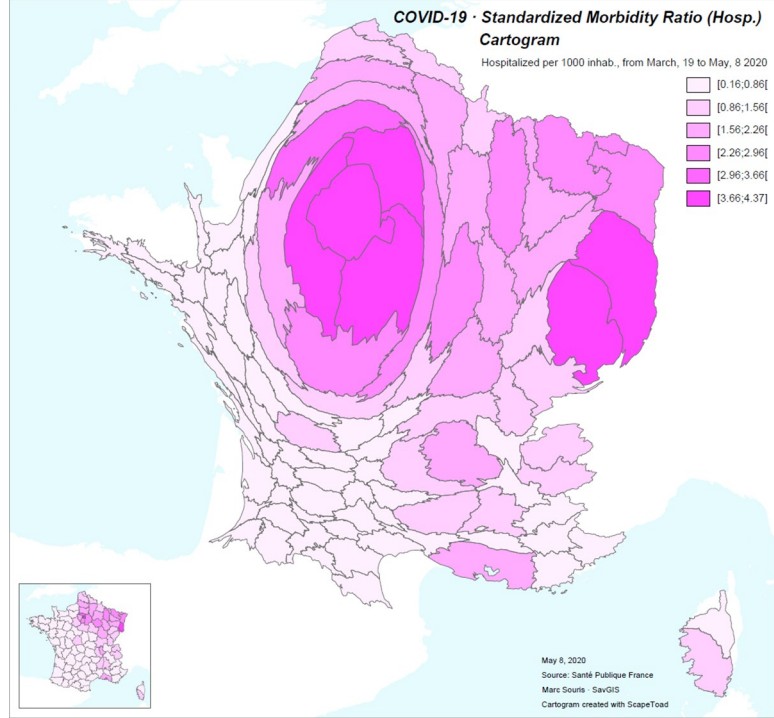

**Fig 5. Cartogram of the standardized hospitalization rate for COVID-19 in France from March 19 to May 8, 2020.** The geometry of the districts in France is shown in the lower left corner.

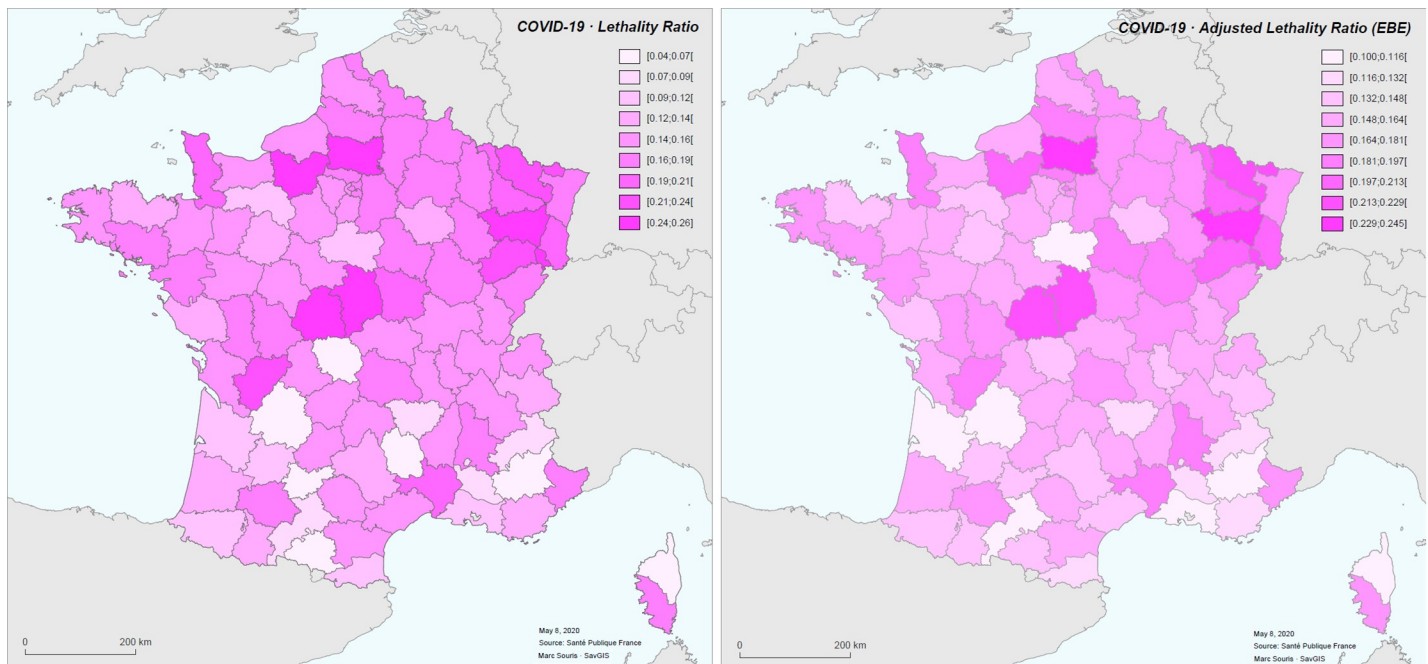

**Fig 6.** Case-fatality rate (left) and case-fatality rate adjusted by empirical Bayesian estimator (right) for the COVID-19 epidemic in France from 19 March to 8 May 2020.

clustering of high case-fatality rate values in regions of high morbidity (particularly the Grand-Est), and shows some cases of districts with high case-fatality rate values isolated in areas with low rates.

The Breslow & Day significance test for SLR shows districts where the SLR is statistically significantly different from 1, corresponding to districts with abnormally high (SLR > 1, red) or abnormally low (SLR < 1, green) standardized case-fatality rates. The individual significance threshold is set at 0.05, and for all districts at 0.0005 to account for multi-testing (Fig 9).

There is no correlation between hospital capacity (bed number per 100,000) and COVID-19 standardized case-fatality rate per district (Bravais-Pearson index r = -0.11).

At district level, there is no significant correlation (Bravais-Pearson index r = 0.14) between case-fatality rate and health risk factors rate (coronary disease, hypertension, acute respiratory

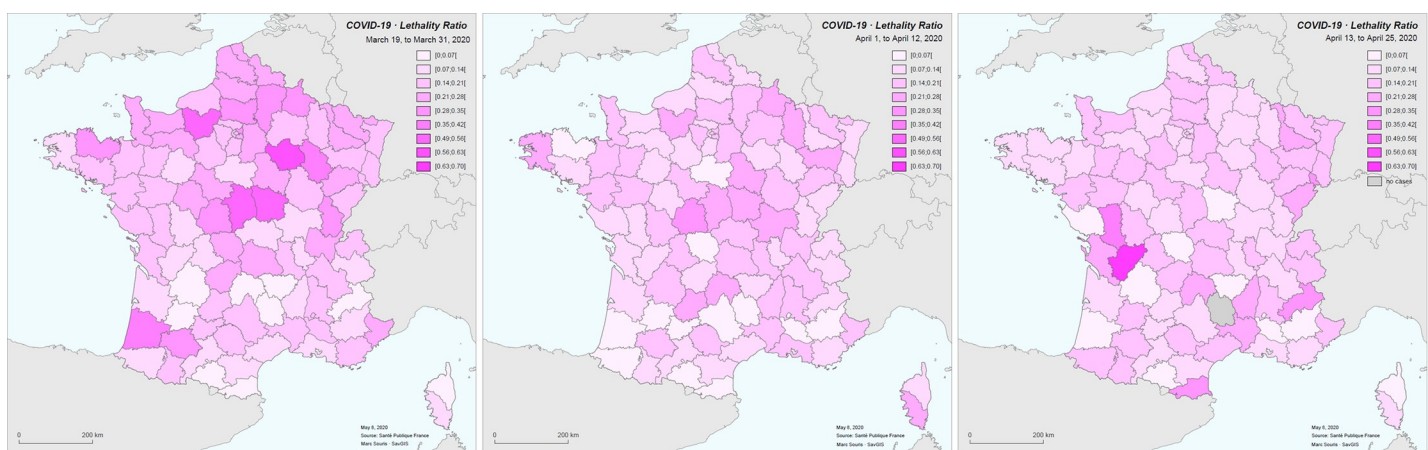

**Fig 7. Calculated case-fatality rates by 13-day period.**

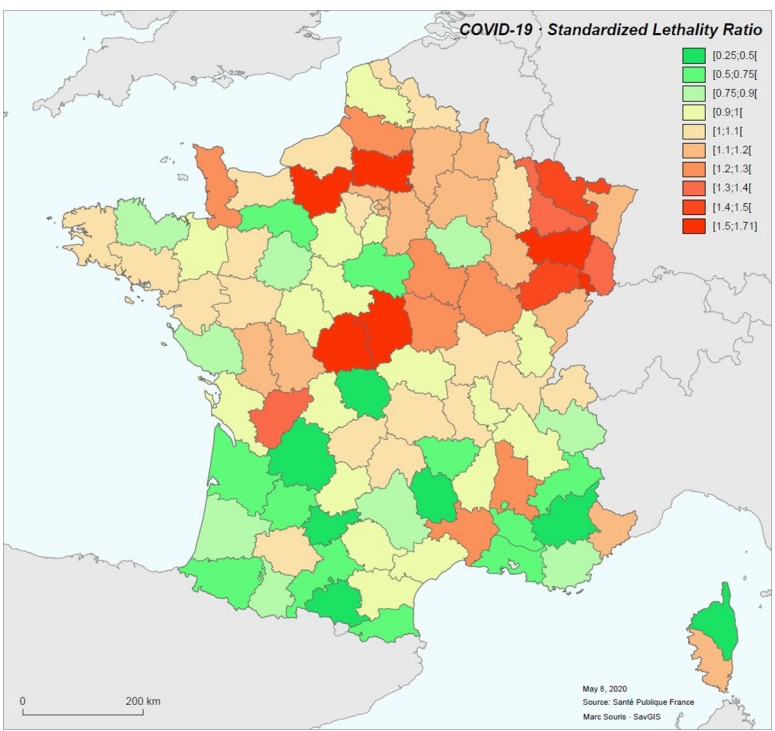

**Fig 8. Standardized Lethality Ratio (SLR) for COVID-19 in metropolitan France for the period March 19 to May 8, 2020.**

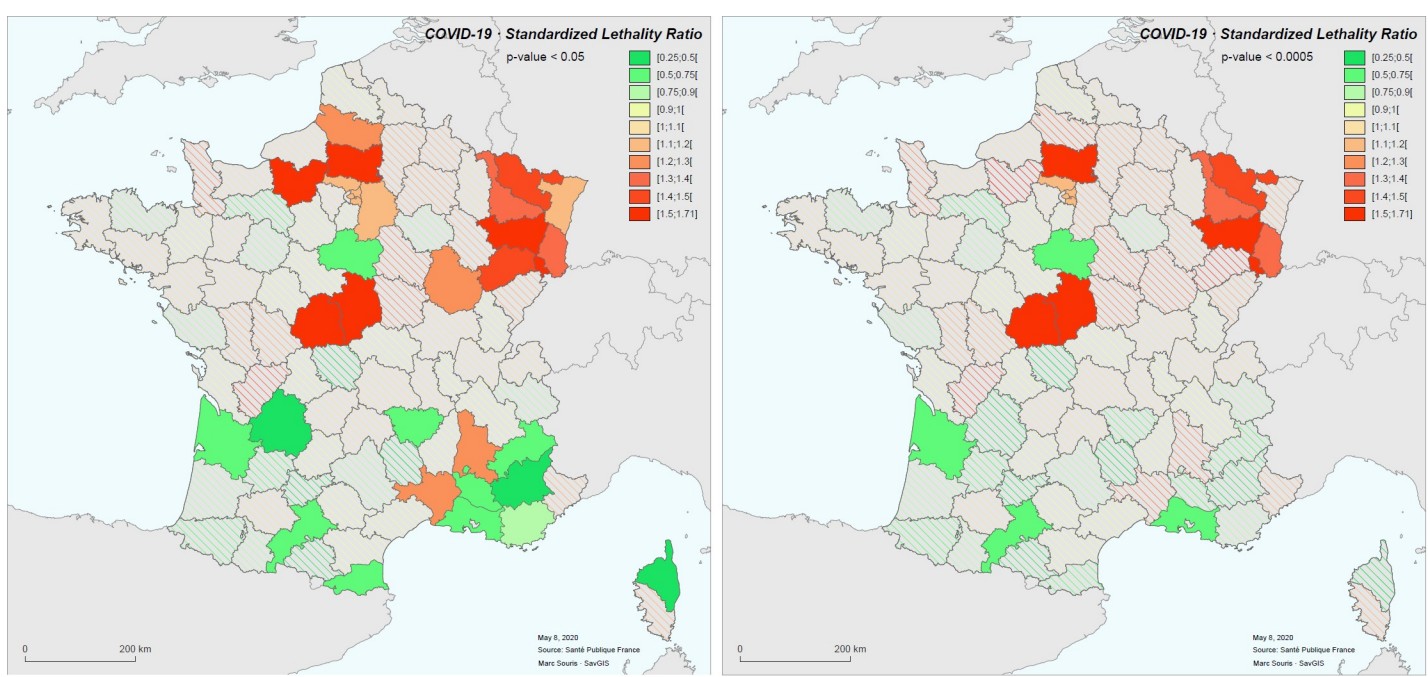

**Fig 9.** Geographical units (districts) with SLR significance: P-value <0.05 (left), p-value < 0.0005 (right).

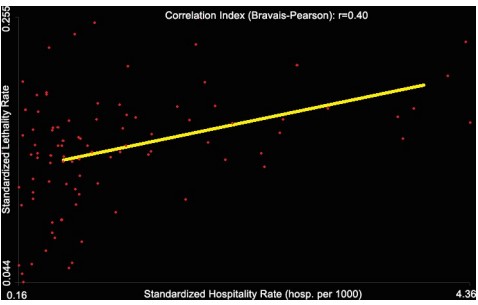

**Fig 10. Correlation between hospitalization rate and CFR during the COVID-19 epidemic in France for the period of time between March 19 to May 8, 2020.**

disease, diabetes). Further investigation with patient data will be necessary to analyze the possible influence of individual health risk factors on SLR at district level

However, there is a significant correlation between the standardized hospitalization rate and the standardized case-fatality rate (Bravais-Pearson index r = 0.40) (Fig 10). The correlation increases (0.48) if we limit the calculation to districts whose SLR is significantly (p-value < 0.05) different from 1.

To illustrate the increase of case-fatality rate with hospitalization rate, Table 5 gives the mean of the standardized case-fatality rate over the districts according to their standardized hospitalization rate. The average case-fatality rate varies from 0.134 for districts with a low hospitalization rate (less than 0.5 per 1000, 31 districts) to 0.152 for all 96 districts. If only districts with high hospitalization rates (above 2 per 1000, 16 districts) are considered, the average case-fatality rate is 0.183. For the eight districts with the highest hospitalization rates ($>$ 3 per 1,000), and considered as district under hospital stress, the average case-fatality rate is 0.191 (Meuse, Moselle, Seine-St-Denis, Hauts-de-Seine, Paris, Val-de-Marne, Haut-Rhin, Territoire de Belfort), while it is 0.149 for all other districts.

A T-test shows that the mean of standardized CFR in France is very significantly (p-value $< 10^{-6}$) higher than the world CFR average, even for districts with low hospitalization rate.

The mapping of the hospitalization rate and the hospital mortality rate with the SLR shows the strong spatial correspondence of these values (Fig 11).

A typology combining hospitalization rates and case-fatality rates is proposed: low rates (values below the mean by less than one standard deviation), high rates (values above the

**Table 5. Average standardized CFR as a function of hospitalization rate during the COVID-19 epidemic in France for the period of time between March 19 to May 8, 2020.**

| Hospitalization rate (per 1000) | Districts number | Standardized CFR (mean) | p-value of T-test with World mean of CFR |
|---|---|---|---|
| < 0.5 | 31 | 0.134 | $< 10^{-6}$ |
| < 1 | 59 | 0.138 | $< 10^{-6}$ |
| < 1.5 | 72 | 0.142 | $< 10^{-6}$ |
| < 2 | 80 | 0.146 | $< 10^{-6}$ |
| < 2.5 | 86 | 0.147 | $< 10^{-6}$ |
| < 3 | 89 | 0.149 | $< 10^{-6}$ |
| < 3.5 | 90 | 0.149 | $< 10^{-6}$ |
| < 4 | 93 | 0.150 | $< 10^{-6}$ |
| All districts | 96 | 0.152 | $< 10^{-6}$ |

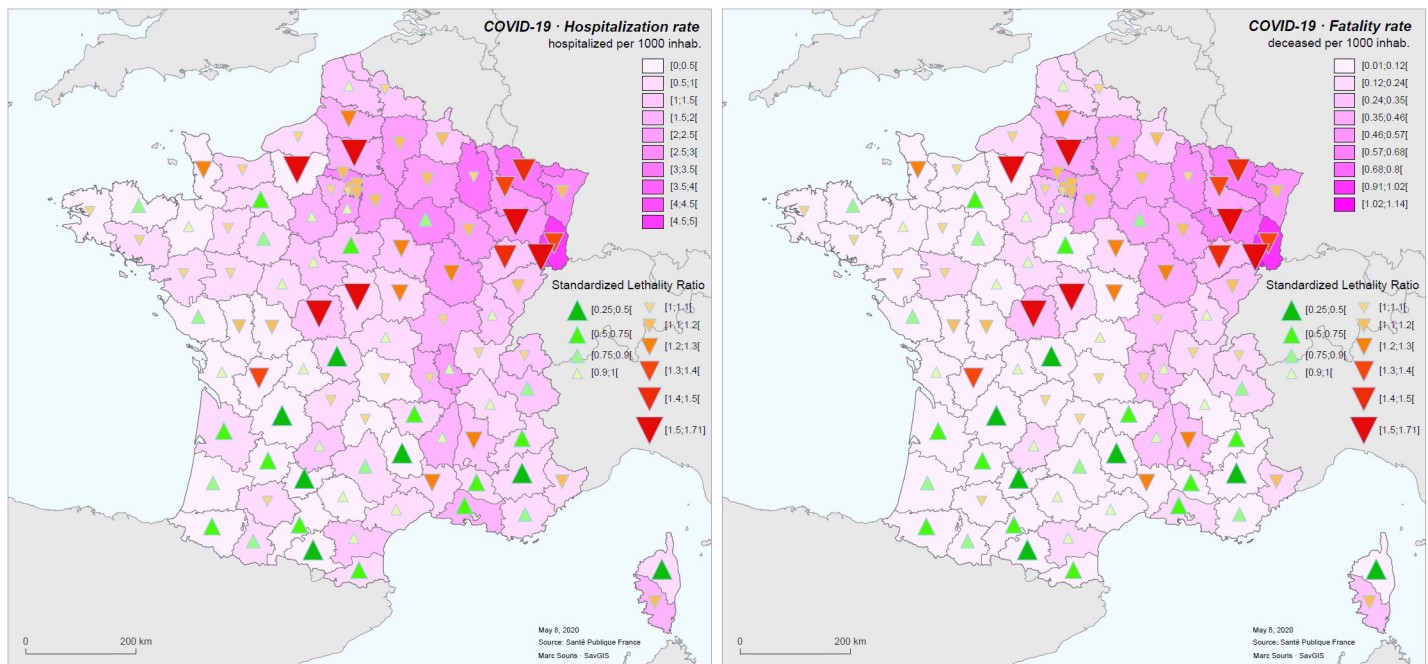

**Fig 11.** SLR (symbols) and hospitalization rate (left) and fatality rate (right) during the COVID-19 epidemic in France for the period of time between March 19 to May 8, 2020.

mean by more than one standard deviation), so as to represent five classes (low-low, low-high, high-low, high-high, other) (Fig 12). The "high hospitalization-high case-fatality rate" class is spatially clustered. The "low hospitalization rate-high lethality rate" class is dispersed, and the population density is significantly lower in that class than in all the other districts (p-value < 0.02).

We call "severity rate" the ratio between the rate of patients in intensive care unit (for reanimation or endotracheal intubation) and the rate of hospitalization. It gives in principle an indication of the severity of the patients in hospital. At district level, the hospitalization rate and this severity rate show a low negative correlation (r = -0.22), indicating a decrease in intensive care rate when the hospitalization rate is high. This trend may be due to the saturation of intensive care units. The relationship between hospitalization and severity could also be interpreted as a decrease in less severe hospitalizations in order to be able to manage more severe cases when the healthcare system is overloaded, which would result in an increase in lethality. Nevertheless, in both cases, there is no correlation between the severity rate and the case-fatality rate (r = -0.10), indicating that globally, the differences in use of resuscitative measures or endotracheal intubation does not impact the case-fatality rate. Finally, the severity rate does not have a spatial distribution corresponding to the spatial distribution of the hospitalization rate (Fig 13).

## Discussion

The standardized case-fatality rates at district level in France (0.04 for Tarn-et-Garonne to 0.26 for the Vosges district) remain in a ratio of 0.3 to 1.6 compared with the national average (mean 0.14, calculated by excluding districts under hospital stress in order not to take account of possible overloading of the healthcare systems). The relationships between standardized morbidity rates and standardized case-fatality rates in France show a correlation between these two indices, with an increase of SLR with hospitalization rate as shown in Table 5. It is

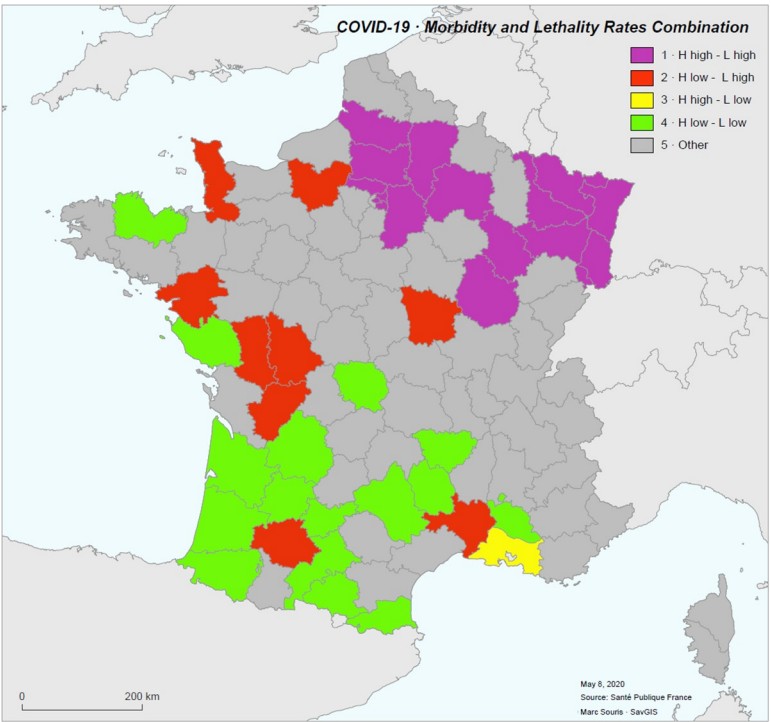

**Fig 12. Combination of standardized hospitalization (H) and case-fatality (L) rates in five classes.**

therefore very likely that high medical admission rates and healthcare workers stress over the period under consideration has increased the hospital case-fatality rate: for the 20 districts with the highest hospitalization rates (essentially located in the Grand-Est and Ile-de-France regions), the average case-fatality rate is 20% higher than the average for all districts, and 25% higher than the average for all other districts alone. It is therefore likely that pressure on hospitals have increased the national average case-fatality rate by district from 0.145 to 0.153. It can thus be estimated that 2,425 deaths (out of the 16,732 deaths due to COVID-19 in hospital in France from 19 March to 8 May 2020, i.e. 15% of the total number of deaths) are due to the overload of the health system in the Grand-Est and Ile-de-France regions.

As we see, after standardization on age, lethality health risk factors (i.e. co-morbidities) do not show, at the district level, any significant correlation with differences in hospital lethality rates. With regard to behavioral risk factors associated to hospital lethality (e.g. late recourse to hospital care), studies focusing on mortality (i.e. not lethality) show significant differences in excess mortality according to the geographical origin of birth of the deceased [16, 17]. The authors of these studies attribute these differences in mortality to differences in the spatial distribution (the foreign-born population is more present in densely populated urban areas, like Ile-de-France region), and not to a risk factors for lethality which could be specific to this population.

There are clearly two risk profiles:

1. the districts where a high rate of hospitalization is coupled with a high case-fatality rate. These districts correspond to the epicenters of the epidemic, are clustered, and include the densely populated areas of Ile-de-France.

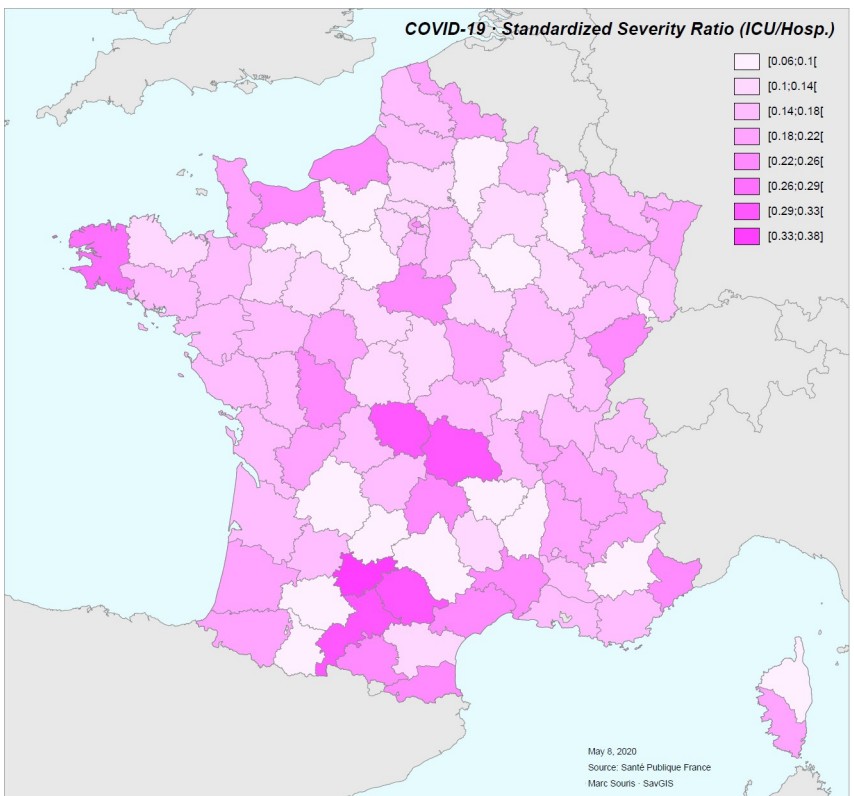

**Fig 13. Severity ratio during the COVID-19 epidemic in France for the period of time between March 19 to May 8, 2020.**

2. the districts where a low rate of hospitalization is coupled with a high case-fatality rate. These districts are geographically dispersed, sparsely populated and mainly located in rural areas.

The first profile accounts for 60% of the patients (59,878 over 99,970). The case-fatality average for this profile is significantly higher (0.187) as compared with the other districts case-fatality average (0.138). The profile displays a 33% excess mortality, corresponding to 2,716 deaths (11,003 observed vs. 8,287 expected with the CFR 0.138). This excess mortality, corresponding to 16% of all deaths, is likely to be the result of increased lethality due to overloading of the healthcare system.

The second profile (more rural) accounts for only 12.5% of the patients (12,514 over 99,970), with an average case-fatality rate significantly high (0.178) corresponding to a 25% excess mortality (2,170 observed vs. 1,731 expected deaths), but of low magnitude. It is possible that this excess mortality is due to a lack of care (medical desert, poor hospital response), which would have led to an increase in hospital lethality.

Some districts in the south of France have both a very low rate of hospitalization and a very low case-fatality rate (Gironde, Dordogne, Gers, Pyrénées Orientales), as a result of a low circulation of the virus and an effective response of the health system. Another particular case, is the one of the Bouches-du-Rhône, which appears with a high hospitalization rate (1.96 per 1,000, for a national average of 1.27) and a low case-fatality rate (0.11 for the Bouches-du-Rhône for a national average per district of 0.15). This situation could be due to a medical policy of more active screening and earlier hospitalization of the infected in the district,

particularly in Marseille metropolitan area. The high value of the hospitalization rate would then be more the result of a non-severe treatment of the infected than of a more active circulation of the virus in the region, as for the lower hospitalization rates in neighboring districts (0.9 for Var, 0.56 for Vaucluse, 0.52 for Gars, 0.86 for Alpes-de-Haute-Provence, 0.81 for Alpes-Maritimes, 0.68 for Hérault). However, patient severity profile (expressed as the ratio between the number of patients admitted to intensive care and the total number of patients hospitalized) is not particularly low in the Bouches-Du-Rhône (0.163 for a mean of 0.156 and a standard deviation of 0.06 for all the districts), and this hospitalization policy is therefore not proven to be the primary factor in the low observed case-fatality rate.

Time analysis shows that the case-fatality rate has decreased over time, globally and in almost all districts, showing an improvement in the management of severe patients during the epidemic. This decrease stabilized rapidly and no longer evolved, in a steady manner. However, this result is limited by the difficulty of calculating a case-fatality rate on aggregated data over a short period of time, with eventual death occurring several days after hospitalization. We chose to delay the calculation of the number of deaths by 12 days compared to the calculation of the number of hospitalizations, since 85% of deaths occur on an average of 13 days after hospitalization [9].

WHO data from 219 countries show a mean case-fatality rate of 0.041 (median 0.029, standard deviation 0.045) with values between 0.31 (Nicaragua) and 0.00 (for countries that did not report deaths until May 8, like Vietnam). Even if we consider only the average case-fatality rate calculated only for the French districts with the lowest hospitalization rates (thus not causing possible saturation of the healthcare system), the French CFR average is very significantly higher than the World average (and the rates of most European countries, such as Spain, 11.73, Greece, 5.52, Germany, 4.28, etc.) (Table 1). The internal variability within France appears too low to explain the large differences in case-fatality rate between countries. These differences cannot be either explained by dissimilarities in morbidity between countries, in lethality risk factors between countries, or inpatient treatment, taking into account the quality of the healthcare system in France (Table 1). One can concluded that the difference between the case-fatality rate calculated with France datasets and the case-fatality rates presented using international WHO data is highly probably the result of a disparity in the registration of cases and/or deaths and not due to the quality of healthcare system or treatment. These differences in the counting of cases and/or deaths may be due to the hospitalization and screening policy specific to each country as well as the ability or willingness to hospitalize more non-severe forms, to the differences in case definition, or to insufficient quality of the health system for detecting and reporting cases and deaths. Such finding is reinforced by studies on excess mortality (difference to the average mortality of previous years over the same period) showing that in most countries, excess mortality is not fully explained by deaths officially attributed to COVID-19, while France datasets present a slight excess of COVID-19 reported deaths [18].

## Conclusion

This study shows that the higher case-fatality rates observed in France in some districts during the first wave of the COVID-19 epidemic (data from 19 March to 8 May 2020) are mostly linked to the level of morbidity in the district, and therefore likely to the congestion of the healthcare systems during the acute phases of the epidemic. When the hospitalization rate is low, high case-fatality rates concern rural districts with low population density and could be linked to healthcare access in these districts.

However, the increase in the standardized case-fatality rate due to exceptional situations during epidemic peaks represents less than 10% of the average case-fatality rate per district in France, and the hospital case-fatality rate without these districts would be reduced from 0.151

to 0.138. This increase cannot therefore explain the extent of the difference observed between the average case-fatality rate in France and the average of the rates reported for all countries by international organizations or information sites (WHO, Wordometer, etc.). These differences probably stem from the reporting of cases and deaths, which is uneven from one country to another, and not from the care or treatment of patients during hospital stress due to epidemic peaks.

## Author Contributions

**Conceptualization:** Marc Souris.

**Data curation:** Marc Souris.

**Formal analysis:** Marc Souris.

**Investigation:** Marc Souris, Jean-Paul Gonzalez.

**Methodology:** Marc Souris.

**Project administration:** Marc Souris, Jean-Paul Gonzalez.

**Software:** Marc Souris.

**Validation:** Marc Souris, Jean-Paul Gonzalez.

**Visualization:** Marc Souris.

**Writing – original draft:** Marc Souris.

**Writing – review & editing:** Jean-Paul Gonzalez.

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
