## [Decision Letter · Decision Letter 0]

30 Jul 2020

PONE-D-20-17129

COVID-19: Spatial Analysis of Hospital Case-Fatality Rate in France

PLOS ONE

Dear Dr. Souris,

Thank you for submitting your manuscript to PLOS ONE. After careful consideration, we feel that it has merit but does not fully meet PLOS ONE’s publication criteria as it currently stands. Therefore, we invite you to submit a revised version of the manuscript that addresses the points raised during the review process.

ACADEMIC EDITOR: I have received the comments of the reviewers on your manuscript. The specific comments of the reviewers are included below. Please provide point by point response in your revised manuscript.

We look forward to receiving your revised manuscript.

Kind regards,

Muhammad Adrish

Academic Editor

PLOS ONE

Journal Requirements:

2.We suggest you thoroughly copyedit your manuscript for language usage, spelling, and grammar. If you do not know anyone who can help you do this, you may wish to consider employing a professional scientific editing service.  

4.We note that [Figure(s) 3, 4, 5, 6, 7, 8, 9, 11, 12 and 13] in your submission contain [map/satellite] images which may be copyrighted. All PLOS content is published under the Creative Commons Attribution License (CC BY 4.0), which means that the manuscript, images, and Supporting Information files will be freely available online, and any third party is permitted to access, download, copy, distribute, and use these materials in any way, even commercially, with proper attribution. For these reasons, we cannot publish previously copyrighted maps or satellite images created using proprietary data, such as Google software (Google Maps, Street View, and Earth). For more information, see our copyright guidelines: http://journals.plos.org/plosone/s/licenses-and-copyright.

1.    You may seek permission from the original copyright holder of Figure(s) [3, 4, 5, 6, 7, 8, 9, 11, 12 and 13] to publish the content specifically under the CC BY 4.0 license. 

Reviewers' comments:

Reviewer's Responses to Questions

**Comments to the Author**

1. Is the manuscript technically sound, and do the data support the conclusions?

Reviewer #1: Partly

Reviewer #2: Yes

Reviewer #3: Partly

2. Has the statistical analysis been performed appropriately and rigorously? 

Reviewer #1: I Don't Know

Reviewer #2: Yes

Reviewer #3: Yes

3. Have the authors made all data underlying the findings in their manuscript fully available?

Reviewer #1: Yes

Reviewer #2: Yes

Reviewer #3: Yes

4. Is the manuscript presented in an intelligible fashion and written in standard English?

Reviewer #1: Yes

Reviewer #2: Yes

Reviewer #3: Yes

5. Review Comments to the Author

Reviewer #1: The manuscript entitled “COVID-19: Spatial Analysis of Hospital Case-Fatality Rate in France” by Souris and Gonzalez has studied the influence of the hospital care system on lethality in metropolitan France during the inception of the COVID-19 epidemic. The study concludes that the higher case fatality rates observed in France is linked to the level of morbidity in the district. The study has also suggested that the case-fatality rates (CFR) may be linked to the hospital care available in the district, i.e., higher congestion of the hospitals or lower accessibility of the hospital care leads to the higher CFR. The manuscript is well written, the introduction is quite informative and the data are well presented and may be interesting for readers from varying fields. However, there are few concerns may be clarified are given below:

1. The abstract does not fully explains the study and needs to mention more about the findings presented in the discussion and conclusion.

2. Higher hospitalization leads to higher CFR, this explains that it is directly related to the higher morbidity, however, to compare with the hospital care system authors need to present a data showing the number of hospital/ beds accessible (per 1000). As explained higher hospital stress means 3 or more per 1000 does not explain the availability or accessibility to hospital care. Table 3 shows the total hospitalized and CFR. Here authors may put a data comparing the hospital care accessibility.

3. Table 3 also should show the data of hospitalized and death (%) 11 and 44 regions. This will help the readers to have better clarity.

4. The higher CFR is among the age group 50 and above. Is the overall difference found between CFR whole and CFR without regions 11 and 44 is due to the presence of higher number of higher age group cases?

5. Table 4 shows that the CFR varies from 0.134 for districts with a low hospitalization rate (less than 0.5 per 1000) to 0.150 for districts with a high hospitalization rate (less than 4 per 1000). How the average for all the districts higher (0.152)? What is the meaning of districts number? What does “all” in the last line shows?

6. Figure 12 presents districts with different morbidity and the lethality. Can the authors compare the low morbidity high lethality vs. high morbidity high lethality districts? Is there any similarity/ differences related to hospitalization/ accessible to hospitals or the aging population in these districts? The CFR is high in the higher age groups. To justify that the CFR is higher in the districts having higher hospital stress or due to lack of hospital care, authors need to show and compare the CFR rates between different age groups. Is the CFR is higher due to the presence of higher %age of higher age groups?

Reviewer #2: Comments to the Author

The authors show the increase in the case-fatality rate in France during epidemic peaks related to the level of morbidity, but not from the care or treatment of patients during hospital stress due to epidemic peaks. Although this paper was well written, I have several comments. Sufficient improvements for those issues are needed for the acceptance to the PLoS One.

Major comments

1. I believe that morbidity and mortality of COVID-19 are strongly related to patient’s factors including incidence of diabetes, and cardiovascular disease. Please indicate the difference of these risk factor between France and other countries or regions in France, or if there is no data, please indicate as a limitation.

2．In our country, there are differences in the availability of infectious disease wards and the capacity to operate ECMO depending on the region. Please explain differences of these factors between France and other countries or regions in France, or if there is no data please explain in the discussions and limitations.

3. Criteria of severity of COVID-19 by admitting ICU or not depend on capacity of each facility, so I concern that it may be inappropriate as an indicator of the severity. I consider that there is no correlation between the severity rate and the case-fatality rate. I believe that endotracheal intubation is relatively good indicator of the severity of the disease. Is it possible to discuss the difference in the severity of cases according to this standard and the difference from other countries?

Reviewer #3: Souris et al present a very unique and interesting analysis of the spatial distribution of COVID-19 cases requiring hospitalization in France, specifically analyzing hospital case-fatality rate to try to understand regional/geographic variations in healthcare. They report significant variations in CFR across regions, and conclude that higher CFR observed by districts are largely related to the level of morbidity/infection rate, and therefore to a putative overwhelming of the local healthcare systems during the acute phases of the epidemic, rather than any deficiency or failure of the local healthcare. Moreover, they suggest the geographic differences are not due to population risk factors or reporting systems, which they purport are uniform across France. Interestingly, the CFR variation is small compared to much greater variation across countries reported in the literature, suggesting marked international inconsistency in case reporting and death ascertainment/coding. The manuscript is very thorough, well-illustrated through figures, with very strong statistical assumptions-based robust analysis, largely well-written, and an important contribution at a time such as this.

Major Comments:

1. Abstract. Final sentence / concluding idea “However, the magnitude of this increase of case-fatality rate represents less than 10 per cent of the average case-fatality rate” is incomplete and leaves the findings unexplained; I would expand and clarify.

2. Data analysis.

a. (Pg 17): “The observed gross hospital case-fatality rate calculated by district showed a mean of 0.153 with a standard deviation of 0.045 … Without the Ile-de-France and Grand-Est regions, the mean gross case-fatality rate per district is 0.149 with a standard deviation of 0.048 (figure 6).” Given minimal change in mean CFR and SD by excluding the hardest hit areas, I don’t understand why the authors persist in the analysis with / without these regions

b. (Pg 18): “The spatial distribution of standardized morbidity rate (hospitalized cases) shows a significant spatial autocorrelation (Moran index: 1.54, p-value < 10-6), and this is expected for an infectious disease.” I don’t understand “autocorrelation” – a parameter with itself, or a value in a region correlating with values around it?

c. (Pg 19): “There is a correlation between the standardized hospitalization rate and the standardized case-fatality rate (Bravais-Pearson index=0.40) (Figure 10), a correlation which increases (0.48) if we limit the calculation to districts whose SLR is significantly different from 1 (p-value < 0.05).” It is unclear which correlation result the p value refers to? Relatedly, p value should be provided in Fig 10 or its legend.

3. Figures/Tables. Some are poorly formatted, with small fonts, poorly explained terminology, and with some sloppiness (French labels from software that are not translated/explained)

a. Fig 1. Y-axis title “Effectifs” unexplained. Why are data in decimal format compared to % format in Table 1?

b. Fig 2. Y-axis titles illegible. X-axis title “Calendar” not clearfor a linear numerical scale.

c. Table 3. Each age group is assigned a % for the various parameters, but the totals are decimal (1.0) !

d. Fig 12. The legend terminology “1-5, H for 1-4” is completely unclear! The wording is clear but does not relate to “H”.

4. Discussion: Several concluding statements are unclear, and don’t seem to be supported by data specifically presented in the Results. Moreover, much new data/analysis/calculations seem to be presented in the Discussion, which is inappropriate.

a. Pg 22: “the average case-fatality rate for all districts being about 15% higher than the average rate calculated in the 20% of districts with the lowest hospitalization rates” makes little sense. Moreover, the following conclusion “It is therefore very likely that the increase in hospital tension over the period under consideration has increased the hospital case-fatality rate” does not follow logically from the preceding.

b. Pg 23: “The first profile accounts for 60% of the patients (59,878 over 99,970). The case-fatality average for this profile is significantly higher (0.187) as compared with the other districts case-fatality average (0.138). The profile displays a 33% excess mortality, corresponding to 2,716 deaths (11,003 observed vs. 8,287 expected with the CFR 0.138). This excess mortality, corresponding to 16 % of all deaths, is likely to be the result of increased lethality due to overloading…” presents much more data/analysis than in the Results. Similarly for the 2nd profile.

c. Pg 24: “WHO data from 219 countries (WHO 20) show a mean case-fatality rate of 0.041 (median 0.029, standard deviation 0.045) with values between 0.31 (Nicaragua) and 0.00 (for countries that did not report deaths, like Vietnam). A T-test shows that the French average is therefore very significantly higher than the world average (p-value < 10-6)”

d. Pg 25: “difference observed between the average case-fatality rate in France and the average of the rates reported for all countries by international organizations or information sites (WHO, Wordometer, etc.). These differences probably stem from the reporting of cases and deaths, which is uneven from one country to another, and not from the care or treatment of patients” appears to be an unsubstantiated conclusion. Because the variation if pandemic stress in some regions of France only explained 10% of the overall CFR variation, why does this exclude the possibility that differences in healthcare quality/access could explain differences across countries?

5. Conclusions. Given the changing population of France with north African and other immigration, could regional population differences not explain some differences in risk of severity of COVID and thus CFR variation? Moreover, regarding the variation in European countries, population and healthcare quality differences are much too easily dismissed: (Pg 10) “In Europe, the characteristics of populations (in terms of risk factor for COVID-19) and health systems are comparable …”

6. References. I am very surprised given the complex geographic and statistical analysis that the reference list is so abbreviated ! It might be helpful to the average reader to perhaps have a few more to support the methodologic approaches, analysis, basics of healthcare statistics, etc.

Minor comments

1. Terminology.

a. Pg 16: “a cartographic representation by cartogram” should be revised.

b. Pg 21: “intensity of reanimation” is unclear.

c. Pg 22: “Hospital tension”

6. PLOS authors have the option to publish the peer review history of their article (what does this mean?). If published, this will include your full peer review and any attached files.

Reviewer #1: No

Reviewer #2: No

Reviewer #3: **Yes: **Prof Sanjay Mehta, MDCM FRCPC

---

## [Author Response · Author response to Decision Letter 0]

14 Aug 2020

Response to the reviewers

PONE-D-20-17129 COVID-19

From: Dr. Marc Souris, Directeur de Recherche, UMR 190 « Unités des virus émergents »

Object: Rebuttal letter to PONE-D-20-17129 COVID-19 review untitled “Spatial Analysis of Hospital Case-Fatality Rate in France” by Marc Souris and Jean-Paul Gonzalez.

To PLOS ONE Editor

Dear Editor,

We thank you to give us the opportunity with our manuscript to fully meet PLOS ONE’s publication criteria. You will find enclosed the revised version of the manuscript that addresses all the points raised by you and the reviewers. As requested, our re-submission includes the following items applied to the Editor’s requirements.

- The present letter as a rebuttal letter responding in detail to each point raised by the academic editor and reviewer(s).

- A marked-up copy of our manuscript with all changes made to the original version (file 'Revised Manuscript D-20-17129 with Track Changes'.

- A copy of our manuscript using track changes as been uploaded as a *supporting information* file.

- An unmarked version of our revised paper without tracked changes (file 'Manuscript D-20-17129 '. (Clean copy of the edited manuscript uploaded as the new *manuscript* file)

- We carefully pay attention for our manuscript to meet PLOS ONE's style requirements, including those for file naming.

- The entire manuscript has been reviewed for language usage, spelling, and grammar by a mother tongue English scientific speaker.

- The abstract has been reviewed and formatted for on the online submission form (via Edit Submission) and identical as the abstract in the manuscript.

- Copyright. We are perfectly aware and confirm that our Figure(s) 3, 4, 5, 6, 7, 8, 9, 11, 12 and 13 do not contain any copyright and our entirely produced by the authors of the present manuscript and will be published by PLOS under the Creative Commons Attribution License (CC BY 4.0) and acknowledge that the manuscript, images, and Supporting Information files will be freely available online, and any third party is permitted to access, download, copy, distribute, and use these materials in any way, even commercially, with proper attribution. 

Please find below our answer addressing point by point all the comments raised by the reviewers.

Reviewer #1: 

1. The manuscript entitled “COVID-19: Spatial Analysis of Hospital Case-Fatality Rate in France” by Souris and Gonzalez has studied the influence of the hospital care system on lethality in metropolitan France during the inception of the COVID-19 epidemic. The study concludes that the higher case fatality rates observed in France is linked to the level of morbidity in the district. The study has also suggested that the case-fatality rates (CFR) may be linked to the hospital care available in the district, i.e., higher congestion of the hospitals or lower accessibility of the hospital care leads to the higher CFR. The manuscript is well written, the introduction is quite informative, and the data are well presented and may be interesting for readers from varying fields. 

Answer: We appreciate such positive comments and thank you for your consideration.

However, there are few concerns may be clarified are given below: 

2. The abstract does not fully explain the study and needs to mention more about the findings presented in the discussion and conclusion.

Answer: The abstract has been completed as requested.

3. Higher hospitalization leads to higher CFR, this explains that it is directly related to the higher morbidity, however, to compare with the hospital care system authors need to present a data showing the number of hospital/ beds accessible (per 1000). As explained higher hospital stress means 3 or more per 1000 does not explain the availability or accessibility to hospital care. Table 4 shows the total hospitalized and CFR. Here authors may put a data comparing the hospital care accessibility.

Answer: 

High hospitalization and therefore high morbidity should not lead to an increase in CFR, however one can think that excessive hospitalization (exceeding reception capacities) will potentially degrade hospital care. 

The healthcare offer is designed to be evenly distributed in France: A table giving the number of beds per 100,000 inhabitants per region has been added, and the source of these data by departments. As it is indicated in the “results” section, there is no direct correlation between lethality and bed density, by department.

Accessibility does not only depend on the provision of care at regional or departmental level, but also depends on many other factors (e.g. socio-economic, behavioral, transport, rurality, etc.).

4. Table 4 also should show the data of hospitalized and death (%) 11 and 44 regions. This will help the readers to have better clarity.

Answer: The CFR for region 11 and 44 only has been added. However, the purpose of the study was to estimate the CFR by age group as a regular basis for the whole country, but not to analyze region 11 and 44 separately.

5. The higher CFR is among the age group 50 and above. Is the overall difference found between CFR whole and CFR without regions 11 and 44 is due to the presence of higher number of higher age group cases?

Answer: No, this is not true when the rate is age-standardized. In fact, the CFR estimate by age group for the whole country is used to standardize the rate on age by district, taking in account the population age structure of each district. 

6. Table 4 shows that the CFR varies from 0.134 for districts with a low hospitalization rate (less than 0.5 per 1000) to 0.150 for districts with a high hospitalization rate (less than 4 per 1000). How the average for all the districts higher (0.152)? What is the meaning of districts number? What does “all” in the last line shows?

Answer: The table shows the increase of CFR with the increase of hospitalization rate. District number represent the cumulative number of districts corresponding to the hospitalization rate indicated in column 1.

7. Figure 12 presents districts with different morbidity and the lethality. Can the authors compare the low morbidity high lethality vs. high morbidity high lethality districts? Is there any similarity/ differences related to hospitalization/ accessible to hospitals or the aging population in these districts? The CFR is high in the higher age groups. To justify that the CFR is higher in the districts having higher hospital stress or due to lack of hospital care, authors need to show and compare the CFR rates between different age groups. Is the CFR is higher due to the presence of higher %age of higher age groups?

Answer: All rates are already age-standardized in order to take into account the difference of the age structure of a population between districts.

Reviewer #2:

8. The authors show the increase in the case-fatality rate in France during epidemic peaks related to the level of morbidity, but not from the care or treatment of patients during hospital stress due to epidemic peaks. Although this paper was well written, I have several comments. Sufficient improvements for those issues are needed for the acceptance to the PLoS One.

Answer: Yes, this true, only the case-fatality rate in France are presented during the epidemic peaks associated to the level of morbidity, by districts, but this is not an analysis done from the care or treatment of patients.

9. I believe that morbidity and mortality of COVID-19 are strongly related to patient’s factors including incidence of diabetes, and cardiovascular disease. Please indicate the difference of these risk factor between France and other countries or regions in France, or if there is no data, please indicate as a limitation.

Answer: In this study, the difference in CFR between districts in France is analyzed, and the magnitude of this difference is evaluated in order to compare it with the difference observed with CFR in other countries. The major observed risk factor for COVID-19 mortality is age, so the data has been age-standardized to exclude this factor as a potential difference between districts. Other risk factors for severity and mortality have been also observed for COVID-19, including among others: hypertension; cardiovascular disease history; diabetes; obesity. When trying to find another risk factor for SLR variation (as hospital congestion for example), it would also be useful to standardize on these health risk factors in order to eliminate differences in risk between districts. Unfortunately, apart for age, we do not have the necessary data to evaluate, for these risk factors or combinations of factors, the populations concerned (by districts) and the global case-fatality rates due to COVID-19 for each factor or a combination of factors. Even if these risk factors are also related to age, this is a limitation, and such appears now in the methods and discussion section.

10. In our country, there are differences in the availability of infectious disease wards and the capacity to operate ECMO depending on the region. Please explain differences of these factors between France and other countries or regions in France, or if there is no data please explain in the discussions and limitations.

Answer: As indicated in the section 8 (Q/A) above, our study analyzes the data reported by district without any clinical data on treatment or patient care at the hospital level. I our knowledge, no specific data on ECMO (Extracorporeal membrane oxygenation) capacity by district is available in France (only at the hospital level and not reported at the national level). By district, no data were available on treatment or care of patients (only for the intensive care units), therefore it was not possible to analyze the differences in lethality by referring to the patient treatment. This kind of analysis can be possible at the hospital level (providing such capacity) and eventually allow analyzing CFR differences between hospitals. But this will be another type of an interesting study.

11. Criteria of severity of COVID-19 by admitting ICU or not depend on capacity of each facility, so I concern that it may be inappropriate as an indicator of the severity. I consider that there is no correlation between the severity rate and the case-fatality rate. I believe that endotracheal intubation is relatively good indicator of the severity of the disease. Is it possible to discuss the difference in the severity of cases according to this standard and the difference from other countries?

Answer: We agree with the comment of the reviewer. In fact, we think that in France ICU admission for COVID patients means most of the time, respiratory distress followed by endotracheal intubation. However, as a criteria of severity of COVID-19 by admitting ICU or not depend not only on the capacity of each facility (some have high capacity and generally before and after the epidemic pic) but also on the criteria (score) used by the ICU team to evaluate the patient (e.g. NEW2) and also depends on the country. Unfortunately, we don’t have access to such type of data for other countries. In this article, we use “severity” based only the rate of resuscitation measures or endotracheal intubation.

Reviewer #3

12. Souris et al present a very unique and interesting analysis of the spatial distribution of COVID-19 cases requiring hospitalization in France, specifically analyzing hospital case-fatality rate to try to understand regional/geographic variations in healthcare. They report significant variations in CFR across regions, and conclude that higher CFR observed by districts are largely related to the level of morbidity/infection rate, and therefore to a putative overwhelming of the local healthcare systems during the acute phases of the epidemic, rather than any deficiency or failure of the local healthcare. Moreover, they suggest the geographic differences are not due to population risk factors or reporting systems, which they purport are uniform across France. Interestingly, the CFR variation is small compared to much greater variation across countries reported in the literature, suggesting marked international inconsistency in case reporting and death ascertainment/coding. The manuscript is very thorough, well-illustrated through figures, with very strong statistical assumptions-based robust analysis, largely well-written, and an important contribution at a time such as this.

Answer: We appreciate the point of view of the reviewer and entirely agree with this overview of our work. Thank you.

13. Abstract. Final sentence / concluding idea “However, the magnitude of this increase of case-fatality rate represents less than 10 per cent of the average case-fatality rate” is incomplete and leaves the findings unexplained; I would expand and clarify

Answer: As above mentioned, the abstract has been entirely revised accordingly to the positive suggestion and comments by the reviewers.

14. Data analysis.

a. (Pg 17): “The observed gross hospital case-fatality rate calculated by district showed a mean of 0.153 with a standard deviation of 0.045 … Without the Ile-de-France and Grand-Est regions, the mean gross case-fatality rate per district is 0.149 with a standard deviation of 0.048 (figure 6).” Given minimal change in mean CFR and SD by excluding the hardest hit areas, I don’t understand why the authors persist in the analysis with / without these regions

Answer: Indeed, the difference is not very significant, but it seemed to us more rigorous to standardize on age with overall rates estimated without a possible increase due to a possible saturation of the care system.

b. (Pg 18): “The spatial distribution of standardized morbidity rate (hospitalized cases) shows a significant spatial autocorrelation (Moran index: 1.54, p-value < 10-6), and this is expected for an infectious disease.” I don’t understand “autocorrelation” – a parameter with itself, or a value in a region correlating with values around it?

Answer: Yes, spatial autocorrelation measures the correlation of a value with the values around it and reflect the spatial pattern of a phenomenon. References has been added.

c. (Pg 19): “There is a correlation between the standardized hospitalization rate and the standardized case-fatality rate (Bravais-Pearson index=0.40) (Figure 10), a correlation which increases (0.48) if we limit the calculation to districts whose SLR is significantly different from 1 (p-value < 0.05).” It is unclear which correlation result the p value refers to? Relatedly, p value should be provided in Fig 10 or its legend.

Answer: The p-value refers to the significance of the SLR to be different from 1, i.e. the district showing statistically significant excess or default of SLR. The correlation can be calculated over all districts, or only over the districts that show statistically significant excess or default lethality.

15. Figures/Tables. Some are poorly formatted, with small fonts, poorly explained terminology, and with some sloppiness (French labels from software that are not translated/explained)

Answer: All French labels has been translated accordingly. All figures and tables have been revised.

a. Fig 1. Y-axis title “Effectifs” unexplained. Why are data in decimal format compared to % format in Table 1?

Answer: Corrected

b. Fig 2. Y-axis titles illegible. X-axis title “Calendar” not clear for a linear numerical scale.

Answer: Corrected

c. Table 3. Each age group is assigned a % for the various parameters, but the totals are decimal (1.0) !

Answer: Corrected

d. Fig 12. The legend terminology “1-5, H for 1-4” is completely unclear! The wording is clear but does not relate to “H”.

Answer: Corrected

16. Discussion: Several concluding statements are unclear, and don’t seem to be supported by data specifically presented in the Results. Moreover, much new data/analysis/calculations seem to be presented in the Discussion, which is inappropriate.

a. Pg 22: “the average case-fatality rate for all districts being about 15% higher than the average rate calculated in the 20% of districts with the lowest hospitalization rates” makes little sense. Moreover, the following conclusion “It is therefore very likely that the increase in hospital tension over the period under consideration has increased the hospital case-fatality rate” does not follow logically from the preceding.

Answer: We reformulate and clarified the sentence.

b. Pg 23: “The first profile accounts for 60% of the patients (59,878 over 99,970). The case-fatality average for this profile is significantly higher (0.187) as compared with the other districts case-fatality average (0.138). The profile displays a 33% excess mortality, corresponding to 2,716 deaths (11,003 observed vs. 8,287 expected with the CFR 0.138). This excess mortality, corresponding to 16 % of all deaths, is likely to be the result of increased lethality due to overloading…” presents much more data/analysis than in the Results. Similarly for the 2nd profile.

Answer: We have chosen to place these results in the discussion because they are part of the analysis of the different profiles we are proposing, which are more a matter of discussion than of results per se.

c. Pg 24: “WHO data from 219 countries (WHO 20) show a mean case-fatality rate of 0.041 (median 0.029, standard deviation 0.045) with values between 0.31 (Nicaragua) and 0.00 (for countries that did not report deaths, like Vietnam). A T-test shows that the French average is therefore very significantly higher than the world average (p-value < 10-6)”

Answer: We moved this paragraph to the “results” section.

d. Pg 25: “difference observed between the average case-fatality rate in France and the average of the rates reported for all countries by international organizations or information sites (WHO, Wordometer, etc.). These differences probably stem from the reporting of cases and deaths, which is uneven from one country to another, and not from the care or treatment of patients” appears to be an unsubstantiated conclusion. Because the variation if pandemic stress in some regions of France only explained 10% of the overall CFR variation, why does this exclude the possibility that differences in healthcare quality/access could explain differences across countries?

Answer: The healthcare system in France is renowned for being of good quality, in terms of equipment, medical staff, patient care and treatment. COVID-19 patient management protocols in France do not differ significantly from those used in other European countries (Germany, Spain, Italy, Greece, etc.). It is therefore unlikely that the huge differences in lethality are due to differences in management or quality of care. But our conclusion concerns the possible degradation of care during periods of hospital stress (epidemic pic, ward and medical worker overload): the results indicate that this observed degradation, even substantial, cannot explain the difference in case-fatality rate with other countries, some of which have themselves experienced periods of hospital stress (e.g. Italy, Spain, UK, Mexico, Brazil).

17. Conclusions. Given the changing population of France with north African and other immigration, could regional population differences not explain some differences in risk of severity of COVID and thus CFR variation? Moreover, regarding the variation in European countries, population and healthcare quality differences are much too easily dismissed: (Pg 10) “In Europe, the characteristics of populations (in terms of risk factor for COVID-19) and health systems are comparable …”.

Answer: Official data collection and statistics on ethnic origin of population are prevented in France. 

18. References. I am very surprised given the complex geographic and statistical analysis that the reference list is so abbreviated! It might be helpful to the average reader to perhaps have a few more to support the methodologic approaches, analysis, basics of healthcare statistics, etc.

Answer: Several references have been added accordingly.

19. Terminology.

a. Pg 16: “a cartographic representation by cartogram” should be revised.

Answer: “Cartogram” is the specific name of this king of map, in which some thematic variable is substituted for land area or distance.

b. Pg 21: “intensity of reanimation” is unclear.

Answer: we change to “the differences in use of resuscitative measures or endotracheal intubation”

c. Pg 22: “Hospital tension”

Answer: we change to “high medical admission rates and healthcare workers stress”

Dear Editor, we thank you for your kind consideration of our reviewed manuscript.

Dr. Marc Souris

Dr. Jean-Paul Gonzalez

---

## [Decision Letter · Decision Letter 1]

1 Sep 2020

PONE-D-20-17129R1

COVID-19: Spatial Analysis of Hospital Case-Fatality Rate in France

PLOS ONE

Dear Dr. Souris,

Thank you for submitting your manuscript to PLOS ONE. After careful consideration, we feel that it has merit but does not fully meet PLOS ONE’s publication criteria as it currently stands. Therefore, we invite you to submit a revised version of the manuscript that addresses the points raised during the review process.

ACADEMIC EDITOR: Please see comments by the reviewers. I am requesting you to address these concerns before I can accept the study for publication.

Please submit your revised manuscript by due date. If you will need more time than this to complete your revisions, please reply to this message or contact the journal office at plosone@plos.org. Please include the following items when submitting your revised manuscript:

We look forward to receiving your revised manuscript.

Kind regards,

Muhammad Adrish

Academic Editor

PLOS ONE

Reviewers' comments:

Reviewer's Responses to Questions

**Comments to the Author**

1. If the authors have adequately addressed your comments raised in a previous round of review and you feel that this manuscript is now acceptable for publication, you may indicate that here to bypass the “Comments to the Author” section, enter your conflict of interest statement in the “Confidential to Editor” section, and submit your "Accept" recommendation.

Reviewer #1: All comments have been addressed

Reviewer #2: All comments have been addressed

Reviewer #3: (No Response)

2. Is the manuscript technically sound, and do the data support the conclusions?

Reviewer #1: Yes

Reviewer #2: Yes

Reviewer #3: Yes

3. Has the statistical analysis been performed appropriately and rigorously? 

Reviewer #1: Yes

Reviewer #2: Yes

Reviewer #3: I Don't Know

4. Have the authors made all data underlying the findings in their manuscript fully available?

Reviewer #1: Yes

Reviewer #2: Yes

Reviewer #3: Yes

5. Is the manuscript presented in an intelligible fashion and written in standard English?

Reviewer #1: Yes

Reviewer #2: Yes

Reviewer #3: Yes

6. Review Comments to the Author

Reviewer #1: The authors have satisfactorily answer all the queries raised during revision. The manuscript now may be considered for publication.

Reviewer #2: The authors show the increase in the case-fatality rate in France during epidemic peaks related to the level of morbidity during hospital stress due to epidemic peaks. Author had responded correctly to my queries.

Reviewer #3: I thank the authors for extensive revisions to the paper, including improving the figures, legends, tables and abstract. The paper reads better. There are however a few issues I raised which were too easily dismissed and perhaps my concern not clear.

Major Comments:

1. My previous comment: d. Pg 25: “difference observed between the average case-fatality rate in France and the average of the rates reported for all countries by international organizations or information sites (WHO, Wordometer, etc.). These differences probably stem from the reporting of cases and deaths, which is uneven from one country to another, and not from the care or treatment of patients” appears to be an unsubstantiated conclusion. Because the variation if pandemic stress in some regions of France only explained 10% of the overall CFR variation, why does this exclude the possibility thatdifferences in healthcare quality/access could explain differences across countries?

Answer: The healthcare system in France is renowned for being of good quality, in terms of equipment, medical staff, patient care and treatment. COVID-19 patient management protocols in France do not differ significantly from those used in other European countries (Germany, Spain, Italy, Greece, etc.). It is therefore unlikely that the huge differences in lethality are due to differences in management or quality of care. But our conclusion concerns the possible degradation of care during periods of hospital stress (epidemic pic, ward and medical worker overload): the results indicate that this observed degradation, even substantial, cannot explain the difference in case-fatality rate with other countries, some of which have themselves experienced periods of hospital stress (e.g. Italy, Spain, UK, Mexico, Brazil).

Unresolved concern: The crux of the paper is that “epidemic stress” leading to some degradation of care explains only a small proportion of the variation in CFR across French regions. However, the authors have not clearly discussed or speculated what is the major reason for this huge variation in lethality! Moreover, their statements of “good quality” of French care and similar patient management protocols across European countries ignores the reality of medical care at the frontline or on the ground, where much variation occurs based on local administrations, local facilities, as well as local populations. Their response is inadequate.

2. My previous comment: Conclusions. Given the changing population of France with north African and other immigration, could regional population differences not explain some differences in risk of severity of COVID and thus CFR variation? Moreover, regarding the variation in European countries, population and healthcare quality differences are much too easily dismissed: (Pg 10) “In Europe, the characteristics of populations (in terms of risk factor for COVID-19) and health systems are comparable …”.

Answer: Official data collection and statistics on ethnic origin of population are prevented in France.

Unresolved concern: This limitation should be recognized/stated in the manuscript, but is not an excuse for lack of scientific consideration by the authors of this important basis for some of the regional / country differences. This point should be considered/discussed even in the absence of any specific data being available.

Minor comments

1. My previous comment: Pg 16: “a cartographic representation by cartogram” should be revised.

Answer: “Cartogram” is the specific name of this king of map, in which some thematic variable is substituted for land area or distance.

Unresolved concern: I accept the term is correct, but the language is awkward/repetitious, “… cartographic representation by cartogram” !

7. PLOS authors have the option to publish the peer review history of their article (what does this mean?). If published, this will include your full peer review and any attached files.

Reviewer #1: No

Reviewer #2: **Yes: **Masaki Okamoto

Reviewer #3: **Yes: **Professor Sanjay Mehta, MDCM FRCPC

---

## [Author Response · Author response to Decision Letter 1]

8 Sep 2020

Please find below our answer addressing the comments raised by the reviewer.

Reviewer #3

R#3- Unresolved concern: The crux of the paper is that “epidemic stress” leading to some degradation of care explains only a small proportion of the variation in CFR across French regions. However, the authors have not clearly discussed or speculated what is the major reason for this huge variation in lethality! Moreover, their statements of “good quality” of French care and similar patient management protocols across European countries ignores the reality of medical care at the frontline or on the ground, where much variation occurs based on local administrations, local facilities, as well as local populations. Their response is inadequate.

Answer: 

When we refer strictly to the data, the standard deviation of the case-fatality rate between French departments is 4.5%, for a case-fatality rate that averages 15.3%. Our results show that these variations are directly related to the level of morbidity in the department, and cannot be explained by possible differences between departments in the COVID-19 assumed risk factors for lethality (age, cardiovascular history, hypertension, diabetes, obesity). 

Independently of the causes of the internal spatial variability of lethality in France, we therefore find that this internal spatial variability is too low to explain the significant differences with the rates published for many other countries: the differences in mean lethality between countries cannot be explained either by the differences in morbidity between countries, or by the differences in lethality risk factors between countries, or by the differences in inpatient treatment (at the global mean level. We certainly do not ignore the existing disparities in the reality of the field, but in this study, it is not a question of going into detail at the granular level of the hospital).

As we write in the article, the most likely explanation for these strong differences in lethality between countries is therefore the disparities in the reporting of cases and deaths. This explanation is confirmed with the publication of studies on excess mortality, which show that most countries have a deficit of deaths (not enough COVID-19 deaths reported to explain excess mortality compared to previous years), while some countries, including France and Belgium, have a slight excess of COVID-19 deaths (too many COVID-19 deaths reported compared to excess mortality compared to previous years).

In order to clarify our point, we have added these few remarks to the end of the discussion (p. 21) and added references to the studies mentioned above.

R#3-Unresolved concern: This limitation should be recognized/stated in the manuscript, but is not an excuse for lack of scientific consideration by the authors of this important basis for some of the regional / country differences. This point should be considered/discussed even in the absence of any specific data being available.

Answer :

Indeed, our answer was probably incomplete. 

Beyond the technical difficulty of invoking ethnic differences or standardizing on this variable (due to lack of data), we consider that ethnic, religious, cultural factors are confounding factors (in the epidemiological sense of the term), in that they can influence the COVID-19 risk factors of lethality (e. g. co-morbidities, access to care, late recourse to care) but are not directly risk factors of lethality themselves: data on lethality risk factors related to the health status of individuals are available, and we already had added a paragraph on this issue in the last revision of the manuscript (page 15 in the Results section). 

Recent studies on excess mortality in March-April 2020 show differences (globally over France) according to the geographical origin of birth of the deceased (but beware: it is not known whether the death is attributable to COVID-19). These differences show an excess of excess mortality among individuals born abroad, particularly in Africa and Asia. The authors of these studies attribute this excess of excess mortality due to COVID-19 to the spatial distribution of this population, which is much more present in densely populated areas, particularly in the Ile-de-France region, and which has therefore been proportionally more affected by the epidemic. These studies concern morbidity and mortality, not lethality, and do not suggest any risk factors for lethality specific to this population. No genetic factor specific to ethno-geographical origin has been evoked to be associated with COVID-19 lethality rates. As suggested by the reviewer, we add a paragraph in the discussion to refer to this question (page 20).

R#3 Minor comments -Unresolved concern: I accept the term is correct, but the language is awkward/repetitious, “… cartographic representation by cartogram” !

Answer: It has been adjusted accordingly in the Figure title by deleting the « cartographic » and rewording the sentence.

---

## [Decision Letter · Decision Letter 2]

23 Oct 2020

PONE-D-20-17129R2

COVID-19: Spatial Analysis of Hospital Case-Fatality Rate in France

PLOS ONE

Dear Dr. Souris,

Thank you for submitting your manuscript to PLOS ONE. After careful consideration, we feel that it has merit but does not fully meet PLOS ONE’s publication criteria as it currently stands. Therefore, we invite you to submit a revised version of the manuscript that addresses the points raised during the review process.

ACADEMIC EDITOR: Please review the comments made by the reviewers and provide a point by point response in your revised manuscript. 

We look forward to receiving your revised manuscript.

Kind regards,

Muhammad Adrish

Academic Editor

PLOS ONE

Reviewers' comments:

Reviewer's Responses to Questions

**Comments to the Author**

1. If the authors have adequately addressed your comments raised in a previous round of review and you feel that this manuscript is now acceptable for publication, you may indicate that here to bypass the “Comments to the Author” section, enter your conflict of interest statement in the “Confidential to Editor” section, and submit your "Accept" recommendation.

Reviewer #3: All comments have been addressed

Reviewer #4: (No Response)

2. Is the manuscript technically sound, and do the data support the conclusions?

Reviewer #3: Yes

Reviewer #4: No

3. Has the statistical analysis been performed appropriately and rigorously? 

Reviewer #3: I Don't Know

Reviewer #4: No

4. Have the authors made all data underlying the findings in their manuscript fully available?

Reviewer #3: Yes

Reviewer #4: No

5. Is the manuscript presented in an intelligible fashion and written in standard English?

Reviewer #3: Yes

Reviewer #4: No

6. Review Comments to the Author

Reviewer #3: (No Response)

Reviewer #4: 1) The concrete results in the manuscript include spatial autocorrelation of certain variables, a positive correlation between hospitalization rate and case-fatality rate, and a “weak” negative correlation between hospitalization rate and severity rate. However, there is concrete data and rigorous analysis to support one point the authors tried to stress—non-comparability of case-fatality rates across countries. Therefore, statements on this point should at most be kept in Discussion and removed from Abstract (the 4th paragraph) and Introduction (the 4th paragraph).

2) The exact definition of age standardized case-fatality rate is still unclear. In the formula in page 4, are P[a] and T[a] district specific or not? If they are, there should be subscript i; if not, that means the expectation E is the same in all the district of the France, which is the so-called “average rate for France”? It is unclear whether the district observed case fatality rate was age adjusted or not? My understanding is yes according to the term “district’s SLR”; however, what’s the formula for it? As I pointed out, the only formula so far is not district specific.

3) The language is verbose and many contents are superfluous. I suggest shortening the manuscript, removing unnecessary parts, and making only succinct points. Take an example, the whole page 4 seems can be removed.

4) The last sentence of the second paragraph in Introduction—“…, unlike case-fatality rates, which are normally calculated independently of the number of infected persons”— it is confusing to me, how the case-fatality rate could be independent of the number of infected persons? Isn’t it the denominator?

7. PLOS authors have the option to publish the peer review history of their article (what does this mean?). If published, this will include your full peer review and any attached files.

Reviewer #3: **Yes: **Professor Sanjay Mehta, MDCM FRCPC

Reviewer #4: No

---

## [Author Response · Author response to Decision Letter 2]

23 Oct 2020

Please find below our answers addressing the comments raised by the reviewer#4.

1) The concrete results in the manuscript include spatial autocorrelation of certain variables, a positive correlation between hospitalization rate and case-fatality rate, and a “weak” negative correlation between hospitalization rate and severity rate. However, there is concrete data and rigorous analysis to support one point the authors tried to stress—non-comparability of case-fatality rates across countries. Therefore, statements on this point should at most be kept in Discussion and removed from Abstract (the 4th paragraph) and Introduction (the 4th paragraph).

Answer: we removed the statement of the introduction (effectively it is a result of our study), but we think that it can be kept as a result in the abstract. We move the statement at the end of the abstract to clearly show it as a conclusion.

2) The exact definition of age standardized case-fatality rate is still unclear. In the formula in page 4, are P[a] and T[a] district specific or not? If they are, there should be subscript i; if not, that means the expectation E is the same in all the district of the France, which is the so-called “average rate for France”? It is unclear whether the district observed case fatality rate was age adjusted or not? My understanding is yes according to the term “district’s SLR”; however, what’s the formula for it? As I pointed out, the only formula so far is not district specific.

Answer: Indirect age standardization use case-fatality rates by age group a (Ta) calculated over the whole country, and not district specific, but Pa,i is the population of age group a in the district i. The subscript i was missing for Pa,i. We corrected this error, and we thank the reviewer to have highlighted this error in the formula.

3) The language is verbose and many contents are superfluous. I suggest shortening the manuscript, removing unnecessary parts, and making only succinct points. Take an example, the whole page 4 seems can be removed.

Answer: we agree that the manuscript is long, but we think that explanations on lethality and its different factors are required to better understand the article (as noted positively by the other reviewers).

4) The last sentence of the second paragraph in Introduction—“…, unlike case-fatality rates, which are normally calculated independently of the number of infected persons”— it is confusing to me, how the case-fatality rate could be independent of the number of infected persons? Isn’t it the denominator? 

Answer: The sentence meaning is: the case-fatality rate is independent of the extend of the disease in the population. We removed the sentence to avoid confusion.

5) I confirm that all the data used in this study is fully available, as indicated in the article.

---

## [Decision Letter · Decision Letter 3]

3 Nov 2020

PONE-D-20-17129R3

COVID-19: Spatial Analysis of Hospital Case-Fatality Rate in France

PLOS ONE

Dear Dr. Souris,

Thank you for submitting your manuscript to PLOS ONE. After careful consideration, we feel that it has merit but does not fully meet PLOS ONE’s publication criteria as it currently stands. Therefore, we invite you to submit a revised version of the manuscript that addresses the points raised during the review process.

ACADEMIC EDITOR: Please see below the unaddressed concern raised by the reviewer. I am unable to accept your study until these concerns have been appropriately addressed.

We look forward to receiving your revised manuscript.

Kind regards,

Muhammad Adrish

Academic Editor

PLOS ONE

Reviewers' comments:

Reviewer's Responses to Questions

**Comments to the Author**

1. If the authors have adequately addressed your comments raised in a previous round of review and you feel that this manuscript is now acceptable for publication, you may indicate that here to bypass the “Comments to the Author” section, enter your conflict of interest statement in the “Confidential to Editor” section, and submit your "Accept" recommendation.

Reviewer #4: All comments have been addressed

2. Is the manuscript technically sound, and do the data support the conclusions?

Reviewer #4: Yes

3. Has the statistical analysis been performed appropriately and rigorously? 

Reviewer #4: Yes

4. Have the authors made all data underlying the findings in their manuscript fully available?

Reviewer #4: Yes

5. Is the manuscript presented in an intelligible fashion and written in standard English?

Reviewer #4: Yes

6. Review Comments to the Author

Reviewer #4: 1) The authors currently put in the last sentence of Abstract “…, suggesting marked international inconsistency in case reporting and death ascertainment/coding”, which is exactly what I commented—it is an implication from the results, not the direct results. Note that it is a very strong statement. Therefore, please remove it from Abstract and leave it only in Discussion.

7. PLOS authors have the option to publish the peer review history of their article (what does this mean?). If published, this will include your full peer review and any attached files.

Reviewer #4: No

---

## [Author Response · Author response to Decision Letter 3]

3 Nov 2020

Answer: we removed the statement from the abstract.

---

## [Decision Letter · Decision Letter 4]

25 Nov 2020

COVID-19: Spatial Analysis of Hospital Case-Fatality Rate in France

PONE-D-20-17129R4

Dear Dr. Souris,

We’re pleased to inform you that your manuscript has been judged scientifically suitable for publication and will be formally accepted for publication once it meets all outstanding technical requirements.

Kind regards,

Muhammad Adrish

Academic Editor

PLOS ONE

Additional Editor Comments (optional):

You have satisfactorily answered all queries made by the reviewers

Reviewers' comments:

Reviewer's Responses to Questions

**Comments to the Author**

1. If the authors have adequately addressed your comments raised in a previous round of review and you feel that this manuscript is now acceptable for publication, you may indicate that here to bypass the “Comments to the Author” section, enter your conflict of interest statement in the “Confidential to Editor” section, and submit your "Accept" recommendation.

Reviewer #4: All comments have been addressed

2. Is the manuscript technically sound, and do the data support the conclusions?

Reviewer #4: Yes

3. Has the statistical analysis been performed appropriately and rigorously? 

Reviewer #4: Yes

4. Have the authors made all data underlying the findings in their manuscript fully available?

Reviewer #4: Yes

5. Is the manuscript presented in an intelligible fashion and written in standard English?

Reviewer #4: Yes

6. Review Comments to the Author

Reviewer #4: My comments were all addressed, I have no further critique.

My comments were all addressed, I have no further critique.

7. PLOS authors have the option to publish the peer review history of their article (what does this mean?). If published, this will include your full peer review and any attached files.

Reviewer #4: No

---

## [Editor Report · Acceptance letter]

2 Dec 2020

PONE-D-20-17129R4 

COVID-19: Spatial Analysis of Hospital Case-Fatality Rate in France 

Dear Dr. Souris:

I'm pleased to inform you that your manuscript has been deemed suitable for publication in PLOS ONE. Congratulations! Your manuscript is now with our production department. 

Kind regards, 

on behalf of

Dr. Muhammad Adrish 

Academic Editor

PLOS ONE